# Stable isotopes show *Homo sapiens* dispersed into cold steppes ~45,000 years ago at Ilsenhöhle in Ranis, Germany

Sarah Pederzani [1,2] ✉, Kate Britton[1,3], Manuel Trost[1], Helen Fewlass [1,4], Nicolas Bourgon[1,5], Jeremy McCormack [1,6], Klervia Jaouen[1,7], Holger Dietl[8], Hans-Jürgen Döhle[8], André Kirchner [9], Tobias Lauer[1,10], Mael Le Corre[3,11], Shannon P. McPherron [12], Harald Meller[8], Dorothea Mylopotamitaki[1,13], Jörg Orschiedt [8], Hélène Rougier [14,15], Karen Ruebens [1,13], Tim Schüler [16], Virginie Sinet-Mathiot [1,17], Geoff M. Smith [1,18], Sahra Talamo [1,19], Thomas Tütken [20], Frido Welker [21], Elena I. Zavala [22,23], Marcel Weiss [1,24] & Jean-Jacques Hublin[1,13]

The spread of *Homo sapiens* into new habitats across Eurasia ~45,000 years ago and the concurrent disappearance of Neanderthals represents a critical evolutionary turnover in our species' history. 'Transitional' technocomplexes, such as the Lincombian–Ranisian–Jerzmanowician (LRJ), characterize the European record during this period but their makers and evolutionary significance have long remained unclear. New evidence from Ilsenhöhle in Ranis, Germany, now provides a secure connection of the LRJ to *H. sapiens* remains dated to ~45,000 years ago, making it one of the earliest forays of our species to central Europe. Using many stable isotope records of climate produced from 16 serially sampled equid teeth spanning ~12,500 years of LRJ and Upper Palaeolithic human occupation at Ranis, we review the ability of early humans to adapt to different climate and habitat conditions. Results show that cold climates prevailed across LRJ occupations, with a temperature decrease culminating in a pronounced cold excursion at ~45,000–43,000 cal BP. Directly dated *H. sapiens* remains confirm that humans used the site even during this very cold phase. Together with recent evidence from the Initial Upper Palaeolithic, this demonstrates that humans operated in severe cold conditions during many distinct early dispersals into Europe and suggests pronounced adaptability.

The Middle to Upper Palaeolithic transition marks an important period in human evolutionary history, with the dispersal of *Homo sapiens* across Eurasia and the disappearance of other hominins such as *Homo neanderthalensis* from the fossil record. Archaeological and genetic evidence increasingly demonstrates that this transition involved a complex patchwork of archaeological and biological turnovers including many dispersals of *H. sapiens*[1–5] (but see ref. [6]). To interpret the evolutionary significance of these events, it is crucial to determine the environments and climatic conditions that *H. sapiens* groups encountered during these dispersals.

Prominent models have suggested that early range expansions of *H. sapiens* during the Late Pleistocene were linked to warm climatic

phases that facilitated adaptation to higher latitudes[7,8]. Recent palaeoclimatic data generated directly from archaeological sites have challenged this idea[9,10] but the scarcity of data from a few sites and archaeological technocomplexes means that climatic scenarios of dispersals still lack the complexity that is emerging from the genetic and archaeological records.

Here we add local palaeoclimatic data pertinent to a newly documented early incursion of Late Pleistocene *H. sapiens* into central Europe. We use multiple stable isotope analysis of faunal remains from Ilsenhöhle in Ranis (hereafter, Ranis), Germany, to document the climatic and environmental conditions *H. sapiens* faced during this dispersal, associated with the Lincombian–Ranisian–Jerzmanowician (LRJ) transitional technocomplex[11–13]. Ranis is located in the Orla valley (50° 39.7563′ N, 11° 33.9139′ E, Thuringia, Germany; Extended Data Fig. 1) and is a type site of the LRJ, an archaeological phenomenon of the Middle to Upper Palaeolithic transition, that extends across northern and central Europe. New genetic, proteomic and chronological evidence now links the LRJ with directly dated *H. sapiens* remains at Ranis and documents one of the earliest dispersals of our species into the European continent[13]. Using the isotopic data generated here we demonstrate the climatic and environmental conditions that pioneering *H. sapiens* groups exploited during their initial spread across central and northwestern Europe.

Originally extensively excavated by W. Hülle from 1932 to 1938[14], recent re-excavations from 2016 to 2022 (Thuringian State Office for Preservation of Historical Monuments and Archaeology (TLDA)/Weimar and the Department of Human Evolution at the Max Planck Institute for Evolutionary Anthropology, Leipzig (MPI-EVA) excavations) have allowed a state-of-the-art reassessment of the stratigraphy and chronology of Ranis (Extended Data Fig. 1 and Supplementary Text 1). In the TLDA/MPI-EVA excavations, the LRJ occupations are associated with layers 9 and 8, dating to 47,500–45,800 cal BP and 46,800–43,300 cal BP, respectively, with the main occupation in layer 8 (ref. 13). *H. sapiens* fossil remains were identified in layers 9 (*n* = 1) and 8 (*n* = 3) and the Hülle collection (*n* = 9), with direct [14]C dates of the Hülle specimens matching those of layers 9 and 8 (ref. 13). Zooarchaeological, archaeological and sediment DNA data suggest that the LRJ occupations were ephemeral and low-intensity, with most faunal remains accumulated by carnivores[15]. The LRJ layers are bracketed by Upper Palaeolithic deposits above and potentially Middle Palaeolithic deposits with low artefact density at the base of the sedimentary profile (Extended Data Fig. 1).

Here, we apply oxygen, carbon, nitrogen, strontium and zinc stable isotope analyses to directly [14]C-dated *Equus* sp. teeth (enamel bioapatite and dentine and mandible bone collagen) from the transitional LRJ and Upper Palaeolithic occupations (layers 9–6) to generate evidence of seasonal palaeotemperatures, water availability and changes in vegetation cover experienced by the humans that produced the LRJ record. Fossil teeth were obtained from the Hülle collection (*n* = 14) and the TLDA/MPI-EVA excavation (*n* = 2; Supplementary Table 1). Furthermore, 24 tooth specimens representing a variety of herbivore, omnivore and carnivore taxa were chosen from the Hülle collection for further $\delta^{66}Zn$ and $^{87}Sr/^{86}Sr$ analyses to explore their feeding ecology and mobility (Supplementary Table 1). The stratigraphic layers of the two excavations are clearly correlated and archaeologically equivalent. However, challenges with the documentation of the Hülle faunal collection often prevent clear stratigraphic assignments of faunal specimens (Supplementary Text 1). For this reason, all equid remains studied here were directly [14]C-dated and correlated to the LRJ and *H. sapiens* fossils through the obtained ages.

## Results

### Chronology

Direct radiocarbon dating of 16 equid specimens yielded calibrated [14]C ages ranging from 48,800 to 36,300 cal BP, covering ~12,500 years

(Supplementary Table 2). The radiocarbon dates from the Hülle faunal collection show a large spread of ages for the layers labelled 'brown' (Supplementary Fig. 1). The LRJ grey layer (X) has more consistent ages, ranging from 45,900 to 42,100 cal BP, which overlaps with the date range of the LRJ layer 8 but also with layer 7 of the TLDA/MPI-EVA excavation[13]. Quality control indicators demonstrate exceptional collagen preservation and purity (Supplementary Table 2) and thus this overlap should be attributed to the excavation methodology of the 1930s and the documentation quality of the Hülle faunal collection. To avoid stratigraphic attribution errors, all analyses in the following are made only on the basis of the direct radiocarbon dates of the equid specimens.

### Stable isotope analyses

Oxygen isotope measurements of sequentially sampled tooth enamel phosphate ($\delta^{18}O_{phos}$) show sinusoidal seasonal cycles in all specimens, with high $\delta^{18}O_{phos}$ peaks representing summers and low $\delta^{18}O_{phos}$ troughs representing winter inputs (Supplementary Fig. 2, Supplementary Text 2 and Supplementary Table 3). The $^{87}Sr/^{86}Sr$ values of equids (0.7090–0.7120), undertaken to confirm that $\delta^{18}O$ values are representative of local conditions without bias from long-distance migrations, fall into the range of bioavailable values of Thuringian lithological units and match those observed in hyenas and ursids with typically small to modest home range sizes (Supplementary Text 3, Supplementary Figs. 3 and 4 and Supplementary Table 4). Seasonal $^{87}Sr/^{86}Sr$ intratooth differences are mostly very small (<0.0005), with no systematic changes through time (Supplementary Fig. 3). One individual, R10131, shows a slightly larger seasonal change (0.0008) and also shows a larger seasonal difference in $\delta^{66}Zn$ (Supplementary Figs. 5 and 6), an isotopic tracer that reflects some impacts of underlying bedrock type—in addition to the more prominent dietary effects (Supplementary Text 4). Overlap of $^{87}Sr/^{86}Sr$ values of this specimen with regionally expected values means that long-distance movement remains unlikely but cannot be ruled out entirely. Thus, excluding this specimen from climatic interpretations is the most cautious approach. Furthermore, the correlation between $^{87}Sr/^{86}Sr$ and $\delta^{66}Zn$ seems driven by two outliers and has a very shallow slope, suggesting that $\delta^{66}Zn$ values are predominantly driven by diet (Supplementary Figs. 7 and 8, Supplementary Text 3 and Supplementary Text 4).

Seasonal and mean annual $\delta^{18}O_{phos}$ values show distinct changes over time (Fig. 1 and Supplementary Table 5), starting with mean annual values of ~12–13.5‰ at 48,000–45,000 cal BP, then dipping by ~3‰ to ~9–10‰ at ~45–43 ka cal BP. After this, $\delta^{18}O_{phos}$ values rise back up to a similar level at around 42,500 cal BP, followed by a gap in the data and a phase of high $\delta^{18}O_{phos}$ variability at ~39–36.5 ka cal BP. During the low $\delta^{18}O_{phos}$ excursion at ~45–43 ka cal BP, winter $\delta^{18}O_{phos}$ values fall as low as 9.0‰, while summers only reach 13.1‰ at their highest value. Seasonal amplitudes of $\delta^{18}O_{phos}$ (summer−winter differences) range between 0.9‰ and 4.1‰ overall and are highest at ~45–43 ka cal BP with a mean seasonal amplitude of 3.3 ± 0.8‰. Seasonal $\delta^{18}O_{phos}$ amplitudes are lower in periods of comparatively higher $\delta^{18}O$ and correlate negatively with winter $\delta^{18}O$ ($P$ = 0.0049, $R$ = −0.39, $n$ = 50, Pearson correlation) but not with summer $\delta^{18}O$ (Supplementary Fig. 9). Hence, changes in seasonality are driven predominantly by changes in winter $\delta^{18}O$. The lowest point in $\delta^{18}O_{phos}$ at ~45–43 ka cal BP coincides with the highest dentine and mandible bone collagen $\delta^{15}N$ values of 8.7–6.8‰ (Fig. 1) and the two proxies show a statistically significant correlation, particularly during this time (Supplementary Figs. 10 and 11, for all data $R$ = −0.36, $P$ = 0.045, $n$ = 32; for >42,000 cal BP, $R$ = −0.59, $P$ = 0.012, $n$ = 17, Pearson correlation). After the ~45–43 ka cal BP high point, $\delta^{15}N$ values decline steadily by ~4‰ until they fluctuate between 3.4‰ and 4.5‰.

Carbon stable isotope values of dentine and mandible bone collagen change little over time, with most individuals falling into a range of less than 1‰ (−21.3‰ to −20.6‰; Fig. 1). The $\delta^{13}C$ values do not show any statistically significant correlations with the other isotope systems (Supplementary Fig. 10, Pearson correlation). Across a variety of taxa in

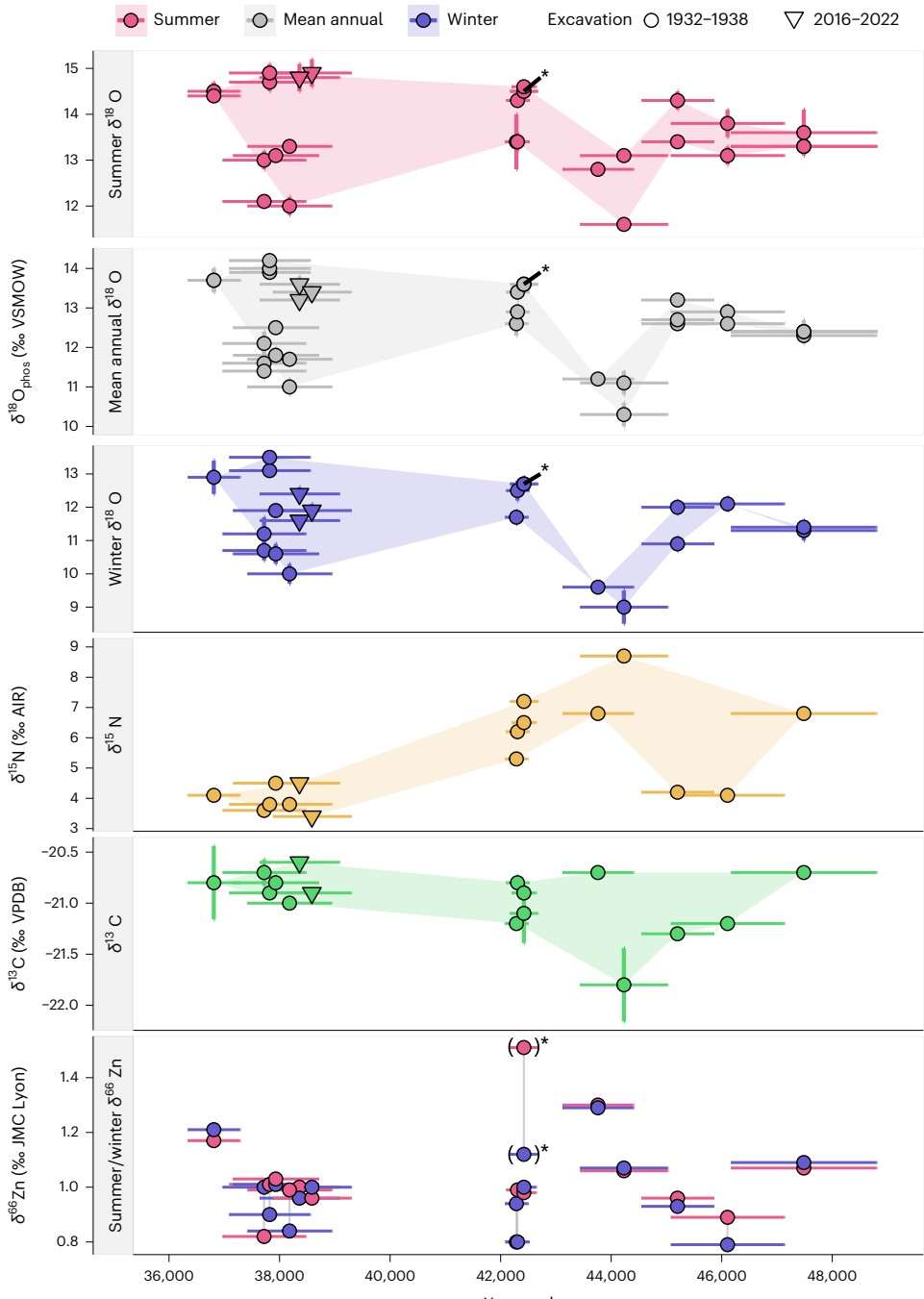

**Fig. 1 | Oxygen, nitrogen, carbon and zinc stable isotope analyses of directly dated equid teeth show changes in climate and environment through the LRJ and Upper Palaeolithic sequence of Ranis.** Summer peak, mean annual and winter trough oxygen isotope values show low values throughout the sequence and a temperature decline from ~48 ka cal BP to a temperature minimum at ~45–43 ka cal BP. This oxygen isotope minimum coincides with high $\delta^{15}N$ (dentine and mandible bone collagen) and $\delta^{66}Zn$ values, suggesting a hypergrazer niche of equids in open steppe environments or very dry soil conditions similarly indicative of an open environment. This is supported by high $\delta^{13}C$ (dentine and mandible bone collagen) values consistent with a steppe or tundra biome. One individual has been marked with an asterisk as it has been excluded from climatic interpretations because $^{87}Sr/^{86}Sr$ and $\delta^{66}Zn$ seasonal amplitudes are high enough that a seasonal movement cannot be completely excluded. Oxygen isotope data points represent $\delta^{18}O$ summer peak, winter trough and annual means of individual annual cycles represented in sinusoidal $\delta^{18}O$ time series

obtained from sequentially sampled tooth enamel (marked in Supplementary Fig. 2). Stable isotope data are presented as the mean ± measurement uncertainty based on sample replicates (1 s.d., $n_{replicates}$ = 3 for $\delta^{18}O$ and $n_{replicates}$ = 2 for all other proxies where error bars are present; replicate measurements represent repeated isotopic measurements of aliquots of each single prepared sample). Measurement uncertainty for $\delta^{66}Zn$ is smaller than the symbol size. Horizontal error bars indicate the 95% calibrated age range of direct radiocarbon dates ($n$ = 1 tooth sample for each data point). Symbol shapes indicate the excavation origin from either the Hülle (1932–1938, circles) or TLDA/MPI-EVA (2016–2022, triangles) campaigns. Collagen analysed for $\delta^{13}C$ and $\delta^{15}N$ was obtained from tooth dentine for all 1932–1938 samples and from adhering mandible bone for the two 2016–2022 samples marked by triangle shapes. Stable isotope delta values are reported in relation to the relevant scale-defining reference materials Vienna Mean Ocean Water (VSMOW), atmospheric $N_2$ (AIR), Vienna Pee Dee Belemnite (VPDB), and Johnson Mattey zinc metal (JMC Lyon).

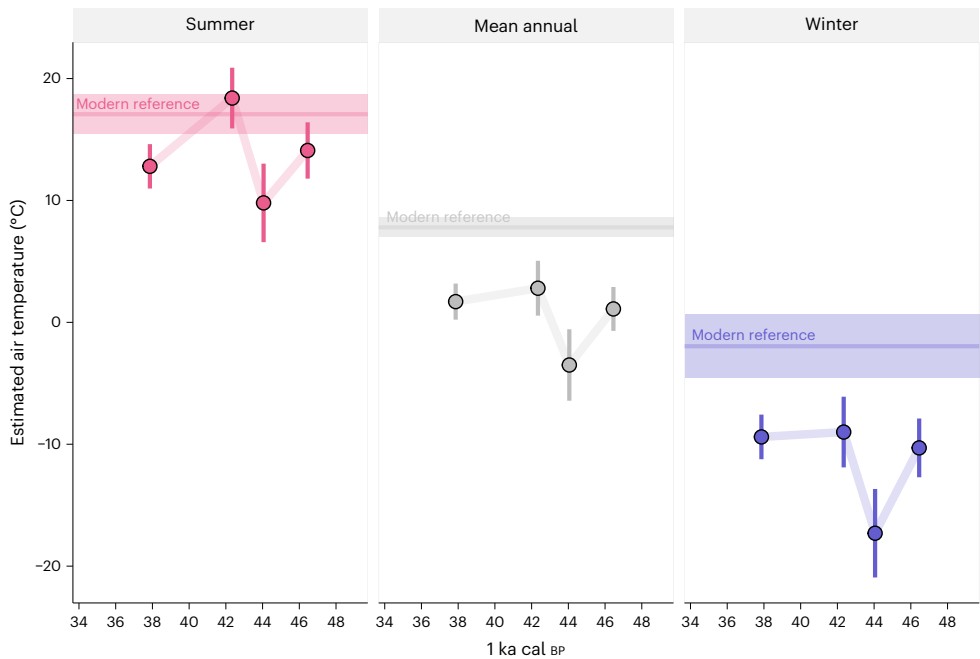

**Fig. 2 | Air temperature estimates derived from $\delta^{18}O$ measurements generally fall below modern-day conditions.** Lowest temperatures are observed in the ~45–43 ka cal BP interval, where they fall ~7–15 °C below modern day and mean annual temperatures below freezing. Oxygen isotope data from several individuals were grouped into time bins according to clusters of radiocarbon dates and $\delta^{18}O$ measurements (Supplementary Table 6). Plotted points represent temperature estimates for each time bin. Error bars represent combined uncertainty for each temperature estimate, taking into account the uncertainty of each temperature calibration step (Supplementary Text 5). $N_{datapoints}$ for each error bar varies by season and time bin and can be found in detail in Supplementary Table 6. In the time bins $\delta^{18}O_{dw}$ estimates are based on a variable number of tooth specimens with $n_{36–39\,ka} = 7$, $n_{42–43\,ka} = 3$, $n_{43–45\,ka} = 2$ and $n_{45–48\,ka} = 3$. Summer and winter temperatures were estimated from inverse modelled $\delta^{18}O$ time series, while annual means were derived directly from unmodelled $\delta^{18}O$ (Supplementary Text 5). Lines and shaded ribbons of modern comparative data represent means and one standard deviation of modern climate observations (MAT, $T_{coldest\,month}$, $T_{warmest\,month}$) for 1961–2009 from the ClimateEU model[59].

the food web, $\delta^{66}Zn$ values of the Ranis fauna follow expected dietary and trophic patterns, with the lowest values observed in carnivores and highest values in herbivores (Supplementary Text 4 and Extended Data Fig. 2). Within herbivores, woolly rhinoceros (*Coelodonta antiquitatis*) show the lowest values, followed by typically browsing or mixed feeding cervid taxa, while equids show the highest $\delta^{66}Zn$ values (0.79‰ to 1.51‰). Across time, $\delta^{66}Zn$ values of equids are highest at ~45–43 ka cal BP, coinciding with the lowest $\delta^{18}O$ values and highest $\delta^{15}N$ values (excluding R10131 ; Fig. 1), resulting in a positive correlation with $\delta^{15}N$ ($R = 0.43$, $P = 0.0013$, $n = 15$, Pearson correlation) and a negative one with $\delta^{18}O$ in the time before 42,000 cal BP ($R = −0.58$, $P = 0.015$, $n = 17$, Pearson correlation; Supplementary Figs. 11 and 12). However, diachronic $\delta^{66}Zn$ change is relatively small and equid $\delta^{66}Zn$ values never overlap with non-herbivore taxa or even some of the lower-$\delta^{66}Zn$ herbivores such as woolly rhinoceros or cervids (Extended Data Fig. 2).

### Water oxygen isotopes and palaeotemperatures

Reconstructed $\delta^{18}O$ values of drinking water ($\delta^{18}O_{dw}$), which enable comparisons with modern meteoric water sources and those of other fauna, fall systematically below modern day $\delta^{18}O$ of local precipitation ($\delta^{18}O_{precip}$), as well as Thuringian rivers and springs (Extended Data Fig. 3 and Supplementary Table 6). Summer $\delta^{18}O_{dw}$ partially overlap with modern spring and river water, as these seasonally more buffered water bodies represent amount-weighted annual averages of $\delta^{18}O_{precip}$. The lowest $\delta^{18}O_{dw}$ values at ~45–43 ka cal BP fall more than 5‰ below the mean annual $\delta^{18}O$ of the modern water sources and more than 8‰ below winter water source values.

This is mirrored by air temperature estimates (Supplementary Table 6), which fall substantially below modern-day conditions with the largest difference in winter (Fig. 2). During the ~45–43 ka cal BP cold phase, air temperature estimates are lower than modern day

by 7.3 ± 3.6 °C in summer, 11.3 ± 3 °C for mean annual conditions and 15.3 ± 4.5 °C in winter. In the oldest data (~48–45 ka cal BP), temperature estimates, while less extreme, still fall 3.0–8.3 °C below modern-day temperatures across all seasons. Temperature seasonality ranges from 22.2 ± 2.6 °C at ~38 ka cal BP to 27.4 ± 3.8 °C during the temperature minimum at ~45–43 ka cal BP, compared to the modern-day temperature seasonality of 19 ± 2.9 °C.

### Relationship with human presence

To test the temporal overlap between the equid specimens yielding the isotopic climate data and the presence of *H. sapiens* we used $\chi^2$ tests and agreement indices of the OxCal Combine function for groups of direct dated equids, *H. sapiens* remains and anthropogenically modified faunal bone fragments. A table of all test results can be found in the associated online supplementary material at https://osf.io/wunfd/.

The direct dates of all equid individuals overlapping with the age ranges of the LRJ deposits (~48–43 ka cal BP) are statistically indistinguishable from at least one directly dated *H. sapiens* fragment and, in many cases, also from at least one anthropogenically modified faunal bone fragment. Importantly, this includes the equids that yielded the lowest $\delta^{18}O$ values (R10124, ETH-111922 and R10126, ETH-111920), which show a calibrated date range of 45,000–43,100 cal BP (Fig. 3). The direct date of R10126 is statistically indistinguishable ($\chi^2 = 0.170$, d.f. = 2, (5% 5.991), Acomb = 117.7) to those of a *H. sapiens* fragment (R10875, ETH-127625) and a cut-marked bone from layer 8 (16/116-159091, ETH-118367), while the date of R10124 is statistically indistinguishable ($\chi^2 = 0.079$, d.f. = 3, (5% 7.815), Acomb = 194.5) from two *H. sapiens* fragments (R10879, ETH-127628 and R10396 ETH-115246) and an equid fragment with percussion notches (16/116-159318, ETH-111935). Other *H. sapiens* specimens and anthropogenically modified

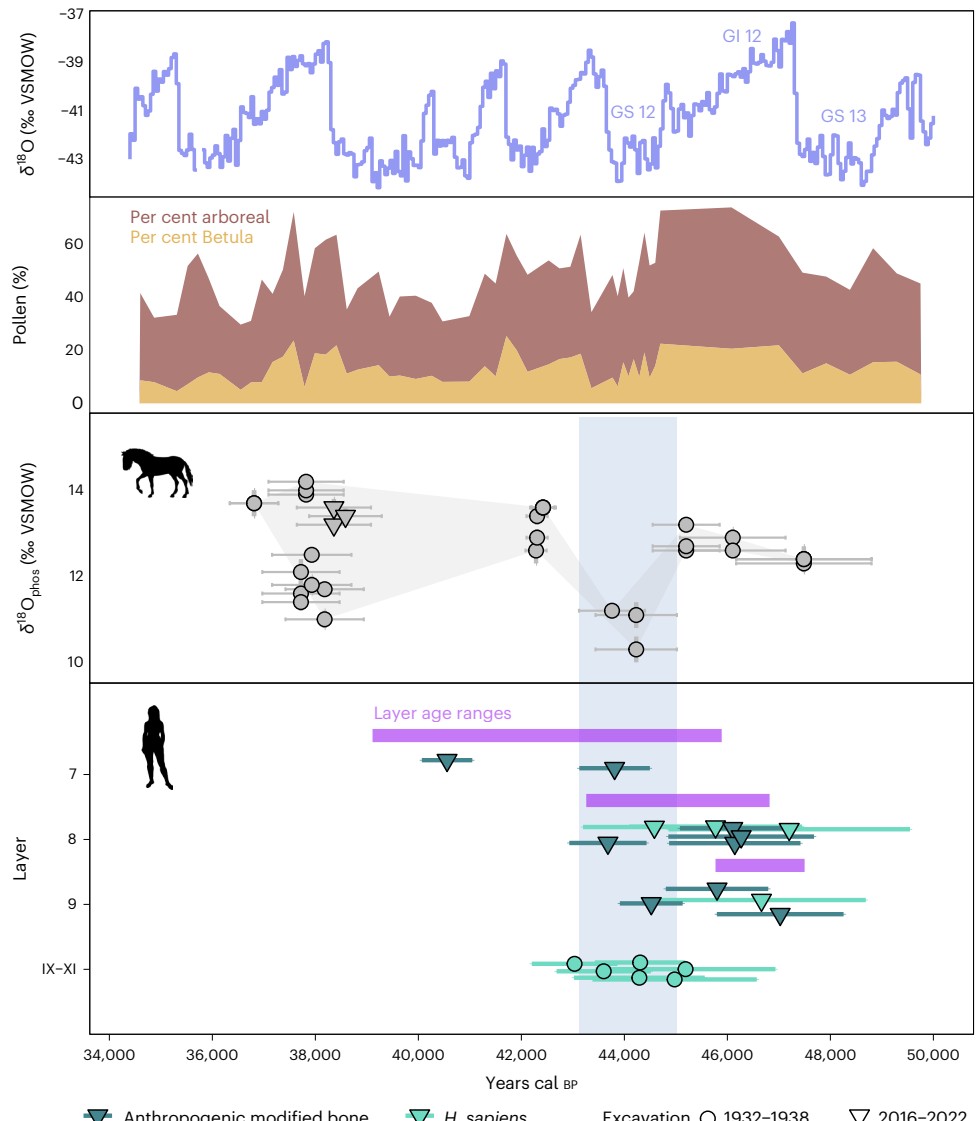

**Fig. 3 | *H. sapiens* presence coincides with the coldest temperatures documented by equid $\delta^{18}$O data.** Comparison of equid $\delta^{18}$O data (top) with directly dated *H. sapiens* remains [13] (bottom, turquoise symbols) demonstrates extensive overlap of *H. sapiens* presence with the coldest temperatures documented between ~45 and 43 ka cal BP (marked by blue shading). This coldest, low $\delta^{18}$O phase overlaps with the age ranges of both the LRJ layer 8 and the beginning of undiagnostic layer 7 (ref. 13) (modelled 95% probability layer age ranges of the MPI-EVA/TLDA excavation in purple) but the direct dates of *H. sapiens* remain and faunal bone fragments with anthropogenic surface modifications clearly show that they endured the cold subarctic steppe conditions evidenced by the stable isotope data at this time. This also holds true independent of the calibration curve, as seen in the uncalibrated dates (Supplementary Fig. 13). Top panels show the relevant Greenland stadials (GS)

and interstadials (GI) recorded in the NGRIP ice cores[60] and the proportions of total arboreal pollen (dark brown) and *Betula* pollen (ochre) in the Füramoos pollen record from southwestern Germany[29,61]. Data are presented as mean ± 95% calibrated age ranges (*n* = 1 bone or tooth enamel sample for each data point), while point shape indicates whether specimens were found in the Hülle (1932–1938, circles) or the TLDA/MPI-EVA (2016–2022, triangles) excavation collection. We argue that *H. sapiens* fragments from the Hülle collection (labelled IX–XI here) all originate from the LRJ deposits (layer X) and were sometimes assigned to a mixture of layer X and adjacent strata by the original excavators due to rough excavation methods (details in ref. 13). We have pooled all these samples here to reflect this. Credits: equid silhouette by Mercedes Yrayzoz, vectorized by T. Michael Keesey (PhyloPic); human silhouette from NASA *Pioneer* plaque.

bone fragments from the LRJ deposits date to a slightly earlier period ~48–45 ka cal BP coinciding with equid specimens yielding temperatures that fall ~3–8 °C lower than today but are less extreme than those from the ~45–43 ka cal BP interval (Fig. 3).

Comparison of the uncalibrated dates confirms that the chronological overlap between *H. sapiens* specimens and the coldest phase at ~45–43 ka cal BP is independent of the calibration curve used (Supplementary Fig. 13). The two equids yielding lowest temperature data (ETH-111922 41,490 ± 360 [14]C BP and ETH-111920 40,740 ± 330 [14]C BP) overlap with the dates of seven of ten human specimens (Supplementary

Fig. 13) and are almost identical to the ages of R10879 (41,429 ± 765 [14]C BP) and R10396 (41,570 ± 420 [14]C BP).

Taxonomic identifications of bone fragments with anthropogenic surface modifications from layers 9 and 8 show that they predominantly originate from reindeer but also include equid fragments. While the total number of fragments (*n* = 12) is too low to make robust inferences about taxonomic representation of hunted herbivores, this does indicate that some equids did overlap with *H. sapiens* occupations and that there is no immediate indication of a systematic difference in the prey taxa targeted by *H. sapiens* compared to carnivores[15].

## Discussion

Using multiple isotope analyses of directly [14]C-dated equid teeth compared to directly dated *H. sapiens* fossils and an updated site chronology, we provide evidence that *H. sapiens* associated with the LRJ occupations of Ranis were present in central Europe during subarctic climatic conditions in a cold open environment, probably including a severe cold episode at ~45–43 ka cal BP. This shows that *H. sapiens* successfully operated in harsh environmental conditions during an early northward range expansion into central Europe.

Equid $\delta^{18}O_{phos}$ data reported here are among the lowest ever reported in Europe for MIS 5 to MIS 3 ($\delta^{18}O_{mean\ annual}$ = 9–13.5‰ for the LRJ). Direct comparisons are limited to data from other equids, where most MIS 3 data from Germany do not fall below 14‰ (ref. 16). Examples with low values from Late Pleistocene stadial contexts in Germany (Bocksteinhöhle, Vogelherdhöhle and Villa Senckendorff[17,18]) and in Switzerland (Boncourt Grand Combe, MIS 3, Courtedoux-Va Tche Tcha, MIS 5a, ref. 19) range from 12.1‰ to 13.2‰ for mean annual data. This is similar to the higher values from Ranis observed for ~48–45 ka cal BP and ~43–39 ka cal BP. Data from Ranis for ~45–43 ka cal BP, however, descend 1.5–2‰ lower than this (Fig. 1). The data from ~45–43 ka cal BP fall ~0.5–1.5‰ lower than even the lowest equid $\delta^{18}O_{phos}$ data reported from the Initial Upper Palaeolithic occupation of Bacho Kiro Cave (minimum $\delta^{18}O_{mean\ annual}$ = 11.2‰), Bulgaria, which has been used to reconstruct subarctic climatic conditions for an early presence of *H. sapiens* in southeastern Europe[9]. Reconstructed drinking water $\delta^{18}O_{dw}$ comparisons with other species suggest closest matches with Late Pleistocene data in Scandinavia and Russia, including data from the Last Glacial Maximum[20,21]. On the basis of characteristics of the study area and on the presence of a sinusoidal signal, we argue that $\delta^{18}O_{phos}$ data reflect palaeotemperatures (Supplementary Text 2), thus our results demonstrate low temperatures, some of them remarkably so. Palaeotemperature estimates show a variability of conditions with a cooling trend but even the comparatively warmer episode at ~48–45 ka cal BP shows mean annual temperatures of <5 °C and based on $\delta^{18}O_{precip}$ most closely matches current climatic conditions in subarctic climate zones in northern Finland (for example, GNIP station Rovaniemi[22]). Mean annual temperatures then descend even further to below freezing for the coldest interval at ~45–43 ka cal BP. For this interval, our results correspond to temperature anomalies of ~7–15 °C below modern-day conditions with largest anomalies in winter (temperature estimation assumptions in Supplementary Text 2). This is paired with a very strong temperature seasonality of up to 27 ± 5 °C, indicating a continental cold subarctic to tundra climate with closest modern-day matches in northwestern Russia (for example, GNIP stations in Amderma and Pechora[22]). Importantly, summer temperature estimates in some cases fall below the 12 °C warmest month isotherm that dictates the Eurasian northern tree line[23], indicating that tree growth may have been impossible in some of the climatic phases captured here. Air temperatures of ~10–15 °C below modern day are consistent with full stadial conditions and have been reconstructed for particularly severe Greenland stadials (GS) in central Europe[24–27].

The cooling trend from ~48 to 44 ka cal BP into full stadial conditions followed by a rapid temperature increase after ~44 ka cal BP matches well with the documented slow cooling and rapid warming of Dansgaard–Oeschger (DO) events and based on tentative correlations with long-term climatic records may capture a Greenland interstadial (GI) to GS transition culminating in a pronounced cold phase such as GS12 or GS13 (refs. 27–30). While an assignment to a specific DO event may not be possible because of the chronometric dating uncertainties involved, the comparison suggests that the equid $\delta^{18}O$ record captures millenial-scale climatic variability that occurred during the time of the LRJ.

Remarkably high $\delta^{15}N$ values of equid dentine and mandible bone collagen in the ~45–43 ka cal BP interval (~7–9‰) compared to earlier and later equid data suggest either a hypergrazer feeding ecology or

dry soil conditions during this interval. Nitrogen isotope variability in arctic biomes shows the highest values in grasses and herbaceous plants over shrubs or trees and this transfers to high $\delta^{15}N$ values in specialized grazers[31]. Glacial phases are often accompanied by low $\delta^{15}N$ values in fauna due to limited nitrogen availability and reduced bacterial activity in cold-wet soils, while high $\delta^{15}N$ values are observed in phases with higher temperatures or low moisture availability[32,33]. As $\delta^{18}O$ values demonstrate low temperatures in this interval, high $\delta^{15}N$ values could indicate dry soil conditions and/or strong grazing specialization of equids, which both imply the presence of an open steppe environment. This matches with reconstructions of grass steppe environments in central Europe during MIS 3 stadials[30,34]. Equid dentine and mandible bone collagen $\delta^{13}C$ values are consistent with feeding in an open grassland environment[35] and the lack of diachronic change is in line with relatively small climatic impacts on $\delta^{13}C$ of C3 plants common to Pleistocene European biomes.

Zinc stable isotope values of the food web at Ranis follow expected trophic level relationships with low values in carnivores and high values in herbivores[36,37]. Within herbivores, $\delta^{66}Zn$ seems to reproduce a pattern of higher values in taxa commonly consuming more grass (equids) and lower values in typical browsers to mixed feeders such as *Cervus elaphus* (Extended Data Fig. 2). A similar pattern has been observed in a few European and African food webs and agrees well with higher $\delta^{66}Zn$ values observed in low-growing plants over higher-standing tree or shrub leaves[37–39] but its robustness is still debated[36] (Supplementary Text 4). While the Ranis data cannot be used to definitively confirm this idea, they are tentatively consistent with it. If true, particularly high and seasonally invariant $\delta^{66}Zn$ observed in equids in the ~45–43 ka cal BP cold interval would support an interpretation of a hypergrazer feeding niche of equids in an open steppe environment. The statistically significant correlation of $\delta^{66}Zn$ with $\delta^{15}N$ (Supplementary Fig. 12) is also consistent with both proxies being driven by grass consumption but due to the effects of soil nutrient cycling on both isotopic systems[40], a relationship with dry soils and consequent changes in soil biochemical cycles is also possible.

Cold temperature conditions and an open grassland or tundra environment for the LRJ at Ranis match the faunal spectrum which includes cold-adapted fauna such as wolverine (*Gulo gulo*), reindeer (*Rangifer tarandus*), woolly mammoth (*Mammuthus primigenius*) and woolly rhinoceros (*Coelodonta antiquitatis*), with reindeer being the predominant herbivore taxon[15]. Furthermore, sedimentological analyses suggest a drop in temperature from layer 9 to the start of layer 7 based on a pronounced decrease in organic carbon and total nitrogen content[13], lending support to the decreasing temperatures from ~48 to 43 ka cal BP reported here.

Our results show that climatic conditions throughout the LRJ occupations, even during the earliest phase ~48–45 ka cal BP, were characterized by temperatures substantially below modern-day conditions. Although a direct contextual connection through anthropogenic modification cannot be established, the chronological overlap between the direct dates of *H. sapiens* remains and anthropogenically modified bone fragments with those of the equid individuals that yielded low temperature results indicates that *H. sapiens* faced subarctic to tundra climatic conditions, probably even those of the severe cold climatic phase 45,000–43,000 cal BP. Zooarchaeological analysis suggests that this presence was characterized by ephemeral occupations, either due to short occupation, task-specific site use or small group sizes[15], although the direct radiocarbon dates of *H. sapiens* remains suggest intermittent site visits across at least a thousand years. Such a site-use pattern, perhaps in the context of frequent movements between sites, may have been a response to the subarctic steppe environment reconstructed for the LRJ. Micromorphological evidence for increased fire use in layer 8 compared to layers 9 or 7 (ref. 13) could also be indicative of a behavioural adaptation to the cooling climate. Owing to the few human modifications on faunal remains, we cannot determine the

seasonality of site occupations by LRJ *H. sapiens* groups[15]. However, more long-term palaeoclimatic records and limited evidence from other LRJ sites (see below) indicate that a subarctic steppe or tundra landscape would have extended over large areas in central Europe, where human groups would have faced similar conditions during potential seasonal movements. The ephemeral *H. sapiens* presence at the site implies that the climatic data probably also cover phases of site formation where humans were absent but we argue that the temporal overlap between directly dated equids, *H. sapiens* remains and anthropogenically modified bones and the reflection of millenial-scale climatic variability in our record does suggest that we can broadly characterize the climates faced by LRJ *H. sapiens* as cold to very cold during most of the LRJ formation period.

The association of *H. sapiens* with the LRJ suggests a rapid range expansion across the northern European plain as far as the British Isles[13], which may have been enabled by the resilience to cold conditions and success in steppe environments documented here. Direct environmental evidence from other LRJ sites is sparse. Nonetheless, the association of LRJ material with cold-adapted fauna at Grange Farm, United Kingdom, and Schmähingen, Germany, as well as biomarker and pollen evidence for climate cooling and open landscapes during the Jerzmanowician occupation of Koziarnia Cave, Poland, suggests that association with cold climatic conditions may be a more common feature of the LRJ than previously noted[41–43]. Genetic data and technological analyses allude to a potential connection between the Ranis LRJ to populations and technocomplexes further east[13], including potentially the Initial Upper Palaeolithic[44], which is associated with a cold-climate *H. sapiens* presence at Bacho Kiro Cave, Bulgaria[9]. Cold-steppe environments that provided open landscapes and supported large herds of prey fauna may have actively supported a rapid dispersal of these connected populations across the northern and eastern European Plain[45,46]. At the same time, our study joins increasing recent evidence for a more complex patchwork of early dispersals of our species in different periods and in more diverse ecological settings than previously appreciated[1–3,5,9,10], raising the question of whether pioneering groups of *H. sapiens* may not be more accurately described as climatically resilient generalists.

## Methods

### Study design, materials and sampling

A multi-isotope study design was chosen to reconstruct a variety of climatic, environmental and ecological aspects of the LRJ and Upper Palaeolithic deposits of Ilsenhöhle in Ranis. A multi-isotope approach also has substantial benefits in reducing equifinality in the interpretation of each isotopic proxy. Oxygen stable isotope analysis was chosen as the main palaeoclimatic proxy, while strontium isotope analysis serves to confirm that sampled animals did not undergo long-distance migrations that could affect the $\delta^{18}O$ signal (Supplementary Texts 2 and 3). Carbon and nitrogen stable isotope analyses were conducted to reconstruct dietary ecology, the structure of the plant biome and water availability in the past. We add to this aspect using zinc stable isotope analysis, a non-traditional stable isotope proxy with potential to elucidate herbivore dietary ecology (Supplementary Text 4).

A total of 16 equid teeth were selected for stable isotope analysis and radiocarbon dating (Supplementary Table 1). Only fully formed and mineralized teeth were considered and first molars (M1) were excluded to prevent the influence of mother's milk consumption on oxygen isotope ratios. Identification of tooth position was achieved with the help of an experienced equid tooth specialist and teeth where an M1 identification could not be confidently excluded were not sampled. Specimens were obtained both from the collections of the 1930s excavation by W. Hülle (sample numbers starting with 'R', $n = 14$) and the 2016–2022 excavation by the TLDA and the MPI-EVA (sample numbers starting with '16/116', $n = 2$). Teeth were chosen to obtain data on the lower part of the depositional sequence from the black layer

(TLDA/MPI-EVA: 6 black, Hülle VIII) downward including the LRJ occupations and adjacent layers (Extended Data Fig. 1) and sequentially sampled to yield subannually resolved stable isotope data (Supplementary Text 5). In the Hülle excavation, this encompasses layers VIII (Schwarze Schicht), IX (Mittlere Braune Schicht), X (Graue Schicht) and XI (Untere Braune Schicht), where layer X is associated with the LRJ technocomplex. These layers correspond to the depositional sequence from layer 6 black to layer 14 in the TLDA/MPI-EVA excavation, where the LRJ is associated with layers 8 and 9 (ref. 13). Layer information in the Hülle faunal collection is recorded using deposit colour (for example, 'Graue Schicht', meaning grey layer) and approximate depth. For the two brown layers identified by Hülle in the lower depositional sequence (IX and XI), these labels can include colour variations (for example, Braune Schicht, Schokobraune Schicht and Rotbraune Schicht), which probably reflect stratigraphic information but also colour differences between site areas, while layer positions (for example, 'mittlere' and 'untere', meaning middle and lower) are almost always omitted. Owing to sloping terrain and compression of deposits by rockfall in some areas of the site[13], depth information is often of limited use in assigning layer designations. Colour and depth descriptions were used to assign layer association as best as possible but all equid specimens were also directly radiocarbon dated using dentine or mandible bone collagen samples to confirm their chronological position.

Because of the predominant role of carnivores in accumulating the faunal remains found in the Ranis LRJ and Upper Palaeolithic deposits and the palimpsest nature of the deposits, the link between faunal stable isotope data and *H. sapiens* activity at the site is less direct than for sites where the faunal assemblage is predominantly anthropogenically accumulated. Moreover, we rely on a comparison of direct radiocarbon dates of the equids analysed for stable isotopes with those of the archaeological layer boundaries, direct dates of *H. sapiens* skeletal remains and anthropogenically modified faunal fragments to establish the archaeological context for the equid remains sourced from the Hülle collection. This approach, while unavoidable due to the characteristics of the site, carries some uncertainty in relating the isotopic climate evidence to periods of *H. sapiens* site occupation. Indeed, it is most likely that the climatic data generated in this study cover both periods of *H. sapiens* presence at Ranis and periods where humans were absent. We use $\chi^2$ tests and agreement indices of the OxCal Combine function to test the chronological agreement between the direct dates of the equid specimens yielding the climatic data on the direct dates of *H. sapiens* remains and anthropogenically modified bones from LRJ contexts to test the probability of a link with *H. sapiens* presence as best as possible.

Samples of ~300–600 mg of dentine or mandible bone were obtained for collagen extraction from tooth roots when available. If roots were not preserved, pieces were cut from lower sections of the tooth crown and tooth enamel was mechanically removed before demineralization. For two specimens (16/116-123510 and 16/116-124286), adhering mandibular bone was available and sampled instead of tooth dentine. Sample pieces were removed using a diamond-coated rotary disk after cleaning of surfaces using air abrasion (Supplementary Text 5).

In addition to the equid specimens chosen for $\delta^{18}O$, $\delta^{13}C$, $\delta^{15}N$, $\delta^{66}Zn$ and $^{87}Sr/^{86}Sr$ analysis, a total of 24 tooth enamel specimens representing a variety of herbivore, omnivore and carnivore taxa were chosen for further $\delta^{66}Zn$ and $^{87}Sr/^{86}Sr$ analyses to explore patterns across the food web (Supplementary Table 1). These specimens were obtained from the Hülle collection from contexts thought to correspond to the brown layer IX. Given the documentation of the finds from the brown layers in the collection described above, this sample probably represents to some degree a mix of specimens of different LRJ and Upper Palaeolithic stratigraphic units (Supplementary Text 1 and Supplementary Fig. 1) dating between ~48 and 36 ka cal BP. While less than ideal, analysis of the directly dated equid specimens shows that diachronic changes in $\delta^{66}Zn$ are too small to affect broader dietary patterns and trophic

relationships in the food web (Supplementary Text 4). Tooth enamel samples from these specimens were obtained either as powder or piece samples in a positive pressure Flowbox following methods described in Supplementary Text 5. In some cases, several teeth from the same mandible were sampled to obtain sufficient tooth enamel.

## Oxygen stable isotope analysis

Tooth enamel powder samples were converted to silver phosphate for oxygen isotope analysis of bioapatite phosphate using digestion with hydrofluoric acid, followed by crash precipitation of silver phosphate[47,48] (Supplementary Text 5). Following recommendations in ref. 49 we did not use an oxidative pretreatment before silver phosphate preparation. Oxygen isotope delta measurements of $Ag_3PO_4$ were conducted in triplicate using a high-temperature elemental analyser (TC/EA) coupled to a Delta V isotope ratio mass spectrometer via a Conflo IV interface (Thermo Fisher Scientific) (Supplementary Text 5). Oxygen isotope delta values were two-point scale normalized to the VSMOW scale using matrix-matched standards calibrated to international reference materials and scale normalization was checked using three separate quality control standards. Details of normalization standards and quality control outcomes can be found in Supplementary Text 5. Average reproducibility of sample replicate measurements was 0.25‰.

## Inverse modelling and palaeotemperature estimation

Before seasonal palaeotemperature estimation an inverse model following ref. 50 was applied to sinusoidal $\delta^{18}O_{phos}$ time series to remove the time averaging and amplitude damping effects caused by the extended nature of tooth enamel mineralization and the sampling procedure (Supplementary Text 5). It should be noted that this inverse model does not account for the successive decrease in tooth growth and mineralization speed that is known to occur in horses towards the completion of tooth formation. Seasonal amplitudes reconstructed here, therefore, probably represent minimum amplitudes (Supplementary Text 5). Following recommendations developed in ref. 9, summer peak and winter trough values were obtained from the inverse modelled $\delta^{18}O$ curves and processed for palaeotemperature estimation, while mean annual temperatures were estimated on the basis of unmodelled annual means. Summer peak and winter trough values were first identified by visual inspection in the original unmodelled $\delta^{18}O$ time series (Supplementary Fig. 2) and corresponding areas on the inverse model outcome were used to yield corrected summer $\delta^{18}O$ and winter $\delta^{18}O$ values. Unmodelled annual means were calculated as the mean of unmodelled summer peak and winter trough values following ref. 9. Detailed information on the modelling procedure is provided in Supplementary Text 5 and we provide all associated code and data, including the parameters used for each model run and the model outcomes for all specimens of this study in the associated online repository at https://osf.io/wunfd/.

Following methods in ref. 51, air temperature estimates were derived via two regression steps using the empirically determined relationships between (1) tooth enamel $\delta^{18}O_{phos}$ and drinking water $\delta^{18}O$ ($\delta^{18}O_{dw}$) and (2) $\delta^{18}O$ of precipitation ($\delta^{18}O_{precip}$) and air temperature. Regression relationships were established using modern calibration datasets of equid tooth enamel $\delta^{18}O$ and drinking water $\delta^{18}O$ as well as temperature and $\delta^{18}O_{precip}$ data from meteorological measurement stations. Details on modern calibration datasets and the conversion procedure are described in Supplementary Text 5. All calculations with the specific conversion equations can be reproduced using the data and code provided in the associated online repository at https://osf.io/wunfd/. Printed conversion equations based on the same data and excel files to conduct equivalent conversions are also available in ref. 9.

To confirm that herbivore $\delta^{18}O$ values reflect climatic influences without ecological or behavioural biases, some studies recommend the use of several taxa from the same archaeological units. Owing to a lack of teeth from other taxa suitable for sequential sampling (for example, large bovids) this was unfortunately not possible in this study (Supplementary Text 2). However, equid $\delta^{18}O_{dw}$ values have been shown to be in excellent agreement with those of other sympatric taxa (Supplementary Text 2).

## Collagen extraction and radiocarbon dating

Collagen was extracted from the equid teeth in the Department of Human Evolution at the MPI-EVA using HCl demineralization, NaOH humic acid removal, gelatinization and ultrafiltration steps, following the protocol in refs. 52,53. (Supplementary Text 5). The suitability of the extracts for dating was assessed on the basis of collagen yield (minimum requirement ~1%) and the elemental values[54], as reported in Supplementary Text 5 and Supplementary Table 2. All extracts were characteristic of well-preserved collagen (Supplementary Table 2) so were submitted for $^{14}C$ dating via accelerator mass spectrometry. Three of 16 collagen extracts were graphitized and dated at the Curt Engelhorn Center for Archaeometry gGmbH (CEZA, laboratory code: MAMS), while the remaining 13 extracts were dated at the Laboratory for Ion Beam Physics at ETH Zurich, Switzerland (laboratory code: ETH; Supplementary Text 5). Aliquots of a background bone (>50,000 BP) were pretreated and dated alongside the equid samples to monitor laboratory-based contamination and were used in the age calculation of the samples. The $^{14}C$ dates were calibrated in OxCal 4.4 (ref. 55) using the IntCal20 calibration curve[56]. Uncalibrated $^{14}C$ dates ($^{14}C$ BP) are reported with their $1\sigma$ error and, in the text, calibrated ranges (cal BP) are reported at the $2\sigma$ range (95% probability). Calibrated ages at the $1\sigma$ range (68% probability) can be found in Supplementary Table 2. All dates have been rounded to the nearest 10 years.

## Carbon and nitrogen stable isotope analysis

Subsamples of collagen extracts were analysed for their elemental composition (%C, %N) and carbon and nitrogen stable isotope composition using a Flash 2000 Organic Elemental Analyser coupled to a Delta XP isotope ratio mass spectrometer via a Conflo III interface (Thermo Fisher Scientific; Supplementary Text 5). Samples were analysed in duplicate and stable isotope delta values were two-point scale normalized using international reference material IAEA-CH-6, IAEA-CH-7, IAEA-N-1 and IAEA-N-2 for $\delta^{13}C$ and $\delta^{15}N$, respectively. Two inhouse quality control standards were used to check scale normalization and evaluate analytical precision. Replicate sample measurements and measurements of the in-house standards indicate a measurement precision of 0.1‰ or better for both $\delta^{13}C$ and $\delta^{15}N$. Details of calibration methods and quality control indicators can be found in Supplementary Text 5. Elemental composition and C/N ratios used for quality checks of collagen integrity are reported in Supplementary Text 5 and Supplementary Table 2.

## Strontium and zinc stable isotope analysis

Zinc and strontium extraction were both conducted on ~10 mg of tooth enamel powder or pieces. In the case of sequentially sampled equid teeth, two subsamples per specimen, representing the summer and winter seasons (based on $\delta^{18}O$ sinusoid peaks and troughs), were selected for $\delta^{66}Zn$ and $^{87}Sr/^{86}Sr$ analysis. Samples were processed using standard acid digestion and column chromatography purification protocols following refs. 37,57,58. (Supplementary Text 5). Isotopic measurements were conducted using a Neptune Multi-Collector Inductively Coupled Plasma Mass Spectrometer (MC-ICPMS, Thermo Fisher Scientific). Procedural blanks and aliquots of quality control standards were processed alongside each batch of samples to quality check wet chemistry and isotope measurement quality. Details of the instrumental setup, scale normalization and quality control indicators can be found in Supplementary Text 5. A subset of samples was analysed in duplicate with an average reproducibility of 0.000008 for $^{87}Sr/^{86}Sr$ and 0.01‰ for $\delta^{66}Zn$.

It has been shown that dental enamel reliably preserves biogenic strontium and zinc isotope values (see Supplementary Text 4 for details

on the preservation of $\delta^{66}$Zn in fossil tooth enamel). As an additional diagenetic check we evaluated isotope measurements against elemental concentration data to confirm the preservation of biogenic isotopic ratios. A lack of relationship between the two in our results indicates that incorporation of diagenetic Zn or Sr is unlikely (Supplementary Figs. 14 and 15).

## Reporting summary

Further information on research design is available in the Nature Portfolio Reporting Summary linked to this article.

## Data availability

All data presented in this study are openly accessible in electronic form in an Open Science Framework repository (https://doi.org/10.17605/OSF.IO/WUNFD) at https://osf.io/wunfd/ and stable isotope data will be deposited in the IsoArch database (https://isoarch.eu/). Data available in the OSF repository include all stable isotope measurements, radiocarbon dates and isotope-derived palaeotemperature estimates.

## Code availability

R code used to conduct the analyses and reproduce the manuscript and Supplementary Information of this study can be accessed in an Open Science Framework repository (https://doi.org/10.17605/OSF.IO/WUNFD) at https://osf.io/wunfd/. This includes code and data to reproduce the oxygen isotope inverse modelling procedure and the palaeotemperature estimations. More detailed documentation and example scripts and files for the inverse model and the temperature estimation can be found at https://github.com/scpederzani/Oxygen_Inverse_Model and https://github.com/scpederzani/Isotope_Temperature_Calibration. The manuscript and Supplementary Information were written using R and Quarto so that figures and analyses can be transparently reproduced using the available code and data. Details of R packages and computing environment can be found in Supplementary Text 5.

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

## Acknowledgements

The re-excavation of Ilsenhöhle in Ranis was conducted by the TLDA and the MPI-EVA. We thank the TLDA and the State Office for Heritage Management and Archaeology Saxony-Anhalt—State Museum of Prehistory (LDA) for the opportunity to study the Ranis faunal material. In particular, we thank R. Hülshoff (LDA) and I. Widany (LDA) for assistance in accessing the LDA collections. We thank M. Kaniecki, L. Klausnitzer, S. Hesse and P. Dittmann (MPI-EVA) for technical assistance during stable isotope and radiocarbon sample preparation. S. Steinbrenner is thanked for technical assistance with thermal conversion elemental analysis isotope ratio mass spectrometry (TC/EA)-IRMS maintenance and EA-IRMS measurements. Thanks are also due to E. Schulz-Kornas (University of Leipzig) for assistance

with identifying equid tooth positions and S. Tüpke (MPI-EVA) for conducting high-resolution photography of equid tooth specimens. We express our gratitude to J. Krause (MPI-EVA) for support during the *H. sapiens* specimen identification work conducted by H.R., J.O. and H.D. The stable isotope work and radiocarbon dating was funded by the Max Planck Society as part of the PhD and postdoctoral project of S.P. and by the German Research Foundation (DFG) as part of the PALÄODIET Project (378496604) awarded to K.J. and T.T. S.P. is supported by a German Academy of Sciences Leopoldina postdoctoral fellowship (LPDS 2021-13). K.B. is supported by a Philip Leverhulme Prize from The Leverhulme Trust (PLP-2019-284). G.M.S. received funding from the European Union Horizon Europe Research and Innovation Programme under Marie Skłodowska–Curie Grant Agreement 101027850. J.M. received funding from the DFG (Project 505905610). D.M. received funding from the European Union's Horizon 2020 research and innovation programme under the Marie Skłodowska-Curie Grant Agreement 861389 - PUSHH.

## Author contributions

The study was designed by S.P., K.B., M.W., S.P.M., J.-J.H., K.J. and T.T. Archaeological excavation was undertaken by M.W., T.S. and S.P.M., who all contributed contextual information. Zooarchaeological and palaeontological analyses were performed by G.M.S. Radiocarbon dating and carbon and nitrogen stable isotope analysis were conducted by H.F. and S.T. Geological and sedimentological analyses were conducted by A.K. and T.L. Characterization and dating of the *H. sapiens* remains was carried out by D.M., H.F., E.I.Z., V.S.M., K.R., F.W., H.R., J.O. and H.D. T.S., H.-J.D., H.D. and H.M. provided study specimens and contextual information. Sampling, sample processing for oxygen stable isotope analysis and TC/EA-IRMS analysis were carried out by S.P. Sampling for strontium and zinc stable isotope analysis were conducted by S.P. and M.T. Sample processing for strontium and zinc stable isotope analysis was carried out by M.T. MC-ICPMS analysis was conducted by N.B. and J.M. Code and data analyses were written and conducted by S.P. M.L.C. contributed to the spatial analysis of strontium stable isotope results. S.P. wrote the paper with input from all authors.

## Funding

## Competing interests

The authors declare no competing interests.

## Additional information

**Extended data** is available for this paper at https://doi.org/10.1038/s41559-023-02318-z.

**Correspondence and requests for materials** should be addressed to Sarah Pederzani.

[1]Department of Human Evolution, Max Planck Institute for Evolutionary Anthropology, Leipzig, Germany. [2]Archaeological Micromorphology and Biomarkers Laboratory (AMBI Lab), Instituto Universitario de Bio-Orgánica Antonio González, Universidad de La Laguna, San Cristóbal de La Laguna, Spain. [3]Department of Archaeology, University of Aberdeen, Aberdeen, UK. [4]Ancient Genomics Lab, The Francis Crick Institute, London, UK. [5]isoTROPIC Research Group, Max Planck Institute for Geoanthropology, Jena, Germany. [6]Institute of Geosciences, Goethe University Frankfurt, Frankfurt, Germany. [7]Géosciences Environnement Toulouse, Observatoire Midi Pyrénées, UMR 5563, CNRS, Toulouse, France. [8]State Office for Heritage Management and Archaeology Saxony-Anhalt—State Museum of Prehistory, Halle, Germany. [9]State Authority for Mining, Energy and Geology of Lower Saxony (LBEG), Hannover, Germany. [10]Terrestrial Sedimentology, Department of Geosciences, University of Tübingen, Tübingen, Germany. [11]CNRS, UMR 7209 Archéozoologie et Archéobotanique—Sociétés, Pratiques et Environnements (MNHN-CNRS), Paris, France. [12]Department of Human Origins, Max Planck Institute for Evolutionary Anthropology, Leipzig, Germany. [13]Chair of Paleoanthropology, CIRB (UMR 7241—U1050), Collège de France, Paris, France. [14]Department of Anthropology, California State University Northridge, Northridge, CA, USA. [15]Department of Archaeogenetics, Max Planck Institute for Evolutionary Anthropology, Leipzig, Germany. [16]Thuringian State Office for the Preservation of Historical Monuments and Archaeology, Weimar, Germany. [17]University of Bordeaux, CNRS, Ministère de la Culture, PACEA, UMR 5199, Pessac, France. [18]School of Anthropology and Conservation, University of Kent, Canterbury, UK. [19]Department of Chemistry G. Ciamician, Alma Mater Studiorum, University of Bologna, Bologna, Italy. [20]Applied and Analytical Palaeontology, Institute of Geosciences, Johannes Gutenberg University, Mainz, Germany. [21]Globe Institute, University of Copenhagen, Copenhagen, Denmark. [22]Department of Evolutionary Genetics, Max Planck Institute for Evolutionary Anthropology, Leipzig, Germany. [23]Department of Molecular and Cell Biology, University of California Berkeley, Berkeley, CA, USA. [24]Friedrich-Alexander-Universität Erlangen-Nürnberg, Institut für Ur- und Frühgeschichte, Erlangen, Germany. ✉e-mail: scpederz@ull.edu.es

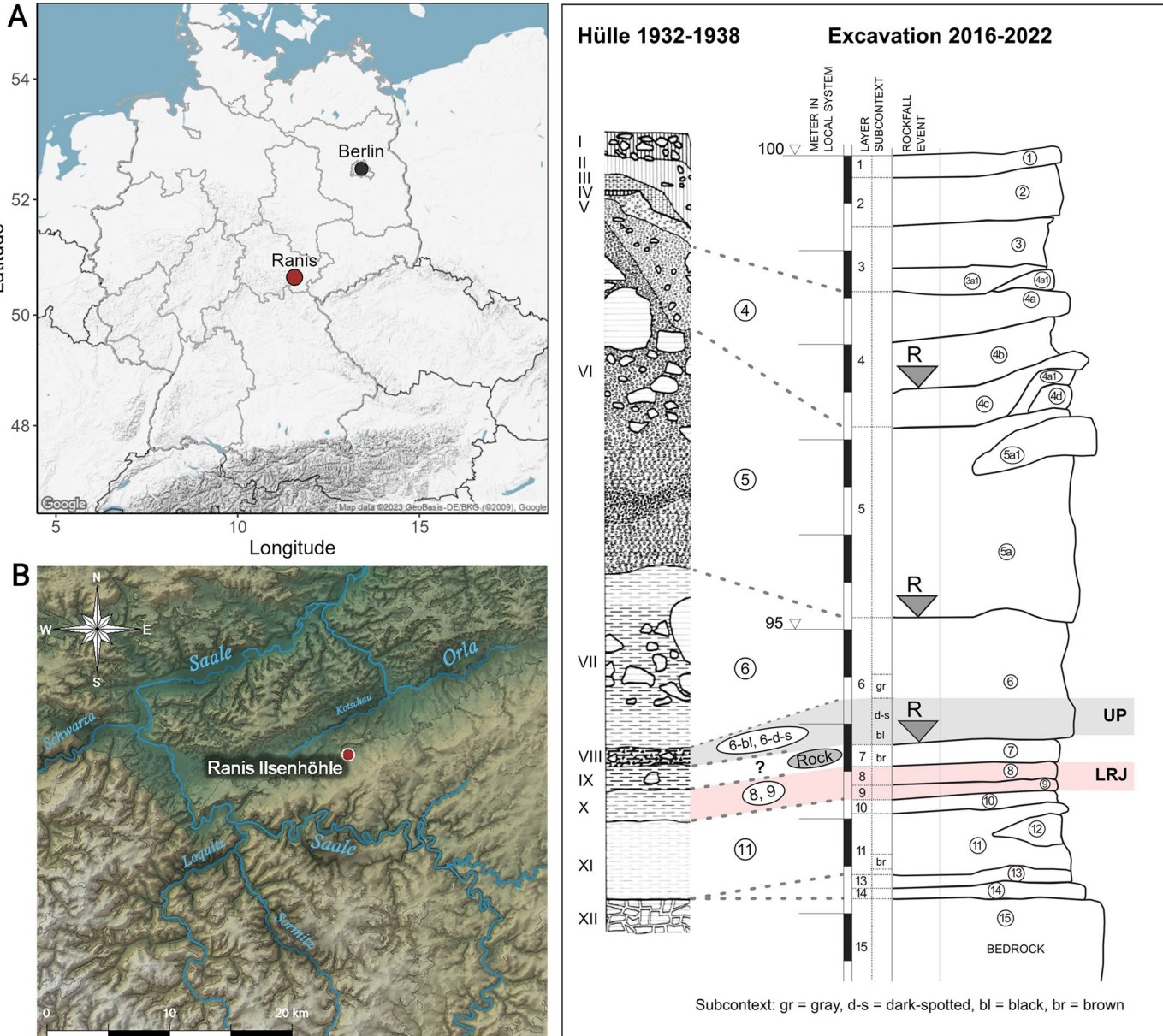

**Extended Data Fig. 1 | Location and stratigraphy of Ilsenhöhle in Ranis, Germany.** A - Location of Ilsenhöhle in Ranis, Thuringia, central Germany. B - Hydrotopographical setting of the area surrounding Ranis in the Orla valley. The Thuringian highlands can be seen to the south of the site. Other notable rivers include the Saale River passing south and west of Ranis. Elevation contour lines are spaced 200 m apart. Elevation data from the European Digital Elevation Model version 1.1[62]. Waterways imported from OpenStreetMap[63]. C - Schematic stratigraphy of the 1930s Hülle excavation and the 2016–2022 TLDA/MPI-EVA excavation with layer correlations (layer numbers in circles). Samples analysed here roughly cover the time period from the Lincombian–Ranisian-Jerzmanowician (LRJ) Layers 8 and 9 marked in red to the Upper Palaeolithic occupation of Layer 6. Rockfall events are marked as 'R'.

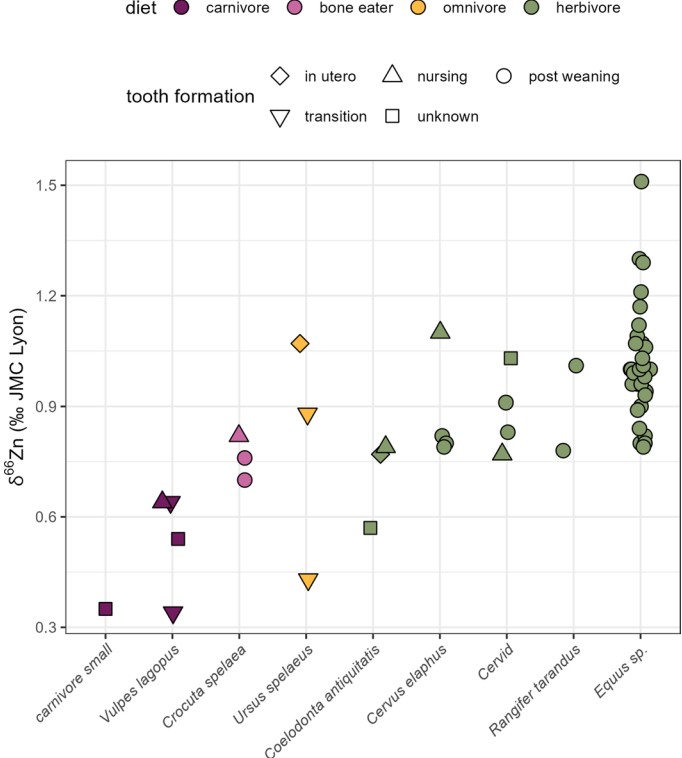

**Extended Data Fig. 2 | Zinc stable isotope values across the Ranis food web.**
Zinc stable isotope ratios of a range of taxa show typical trophic relationships with low $\delta^{66}$Zn values for carnivores, high values for herbivores and intermediate omnivores and bone eating carnivores. Herbivores show a pattern with highest $\delta^{66}$Zn values in equids and lower values in typical browser to mixed feeding cervid taxa such as *Cervus elaphus*. This is potentially consistent with grazer-browser $\delta^{66}$Zn patterns observed in a European Pleistocene food web[38] and a modern African food web[37] and with limited studies of different plant parts[39]. If confirmed, this suggests that higher $\delta^{66}$Zn in some equids could be due to a hypergrazer feeding ecology (see Supplementary Text 4). Shapes indicate stages of tooth development, as teeth formed during nursing or *in utero* can exhibit higher $\delta^{66}$Zn values (see Supplementary Text 4). It should be noted that for equids summer and winter $\delta^{66}$Zn values are plotted (2 per tooth), while other taxa are represented by one measurement per specimen.

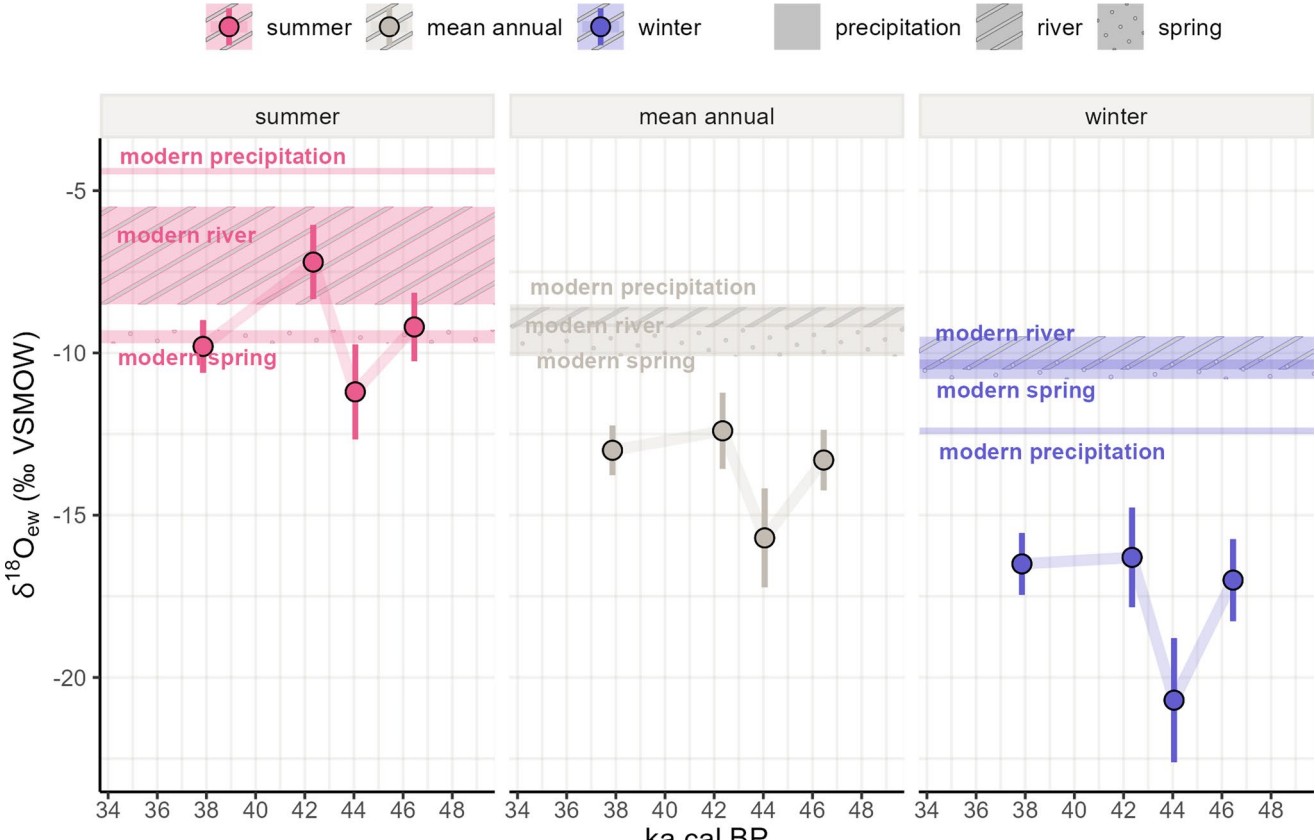

**Extended Data Fig. 3 | Summer, winter and mean annual estimates of environmental water oxygen isotope values compared to modern meteoric water sources.** Reconstructed oxygen isotope composition of drinking water ($\delta^{18}O_{dw}$) fall substantially below $\delta^{18}O$ of modern-day water sources (precipitation - solid ribbons; rivers - hatched ribbons; springs - dotted ribbons), particularly for mean annual and winter values. This indicates that temperatures were substantially below modern-day conditions. Summer $\delta^{18}O_{dw}$ reconstructions partially overlap with modern spring $\delta^{18}O$ due to the pronounced seasonal buffering in groundwaters. Precipitation $\delta^{18}O$ were obtained from estimates for the site location made using the OIPC[64]). River $\delta^{18}O$ data includes measurements from the Heiderbach, a small stream in the Rinne valley[65], the Bode, a Saale tributary in northern Thuringia[66] and the Elbe close to the confluence of the Saale[67]. Spring water $\delta^{18}O$ data includes measurements from four small springs in the Rinne valley[65]. Error bars represent the overall uncertainty introduced by the conversion to drinking water oxygen isotope values (see Supplementary Text 5). $N_{datapoints}$ for each error bar varies by season and time bin and can be found in detail in Supplementary Table 6. In the time bins $\delta^{18}O_{dw}$ estimates are based on a variable number of tooth specimens with $n_{36-39\,ka} = 7$, $n_{42-43\,ka} = 3$, $n_{43-45\,ka} = 2$, $n_{45-48\,ka} = 3$.

# Reporting Summary

## Statistics

For all statistical analyses, confirm that the following items are present in the figure legend, table legend, main text, or Methods section.

| n/a | Confirmed | |
|---|---|---|
| ☐ | ☒ | The exact sample size (*n*) for each experimental group/condition, given as a discrete number and unit of measurement |
| ☐ | ☒ | A statement on whether measurements were taken from distinct samples or whether the same sample was measured repeatedly |
| ☐ | ☒ | The statistical test(s) used AND whether they are one- or two-sided *Only common tests should be described solely by name; describe more complex techniques in the Methods section.* |
| ☒ | ☐ | A description of all covariates tested |
| ☐ | ☒ | A description of any assumptions or corrections, such as tests of normality and adjustment for multiple comparisons |
| ☐ | ☒ | A full description of the statistical parameters including central tendency (e.g. means) or other basic estimates (e.g. regression coefficient) AND variation (e.g. standard deviation) or associated estimates of uncertainty (e.g. confidence intervals) |
| ☐ | ☒ | For null hypothesis testing, the test statistic (e.g. *F*, *t*, *r*) with confidence intervals, effect sizes, degrees of freedom and *P* value noted *Give P values as exact values whenever suitable.* |
| ☒ | ☐ | For Bayesian analysis, information on the choice of priors and Markov chain Monte Carlo settings |
| ☒ | ☐ | For hierarchical and complex designs, identification of the appropriate level for tests and full reporting of outcomes |
| ☐ | ☒ | Estimates of effect sizes (e.g. Cohen's *d*, Pearson's *r*), indicating how they were calculated |

*Our web collection on statistics for biologists contains articles on many of the points above.*

## Software and code

Policy information about availability of computer code

| Data collection | During IRMS data collection Isodat 3.0 was used. |
|---|---|
| Data analysis | Data analysis was conducted using R version 4.2.0. All analysis code is available at https://osf.io/wunfd/ |

For manuscripts utilizing custom algorithms or software that are central to the research but not yet described in published literature, software must be made available to editors and reviewers. We strongly encourage code deposition in a community repository (e.g. GitHub). See the Nature Portfolio guidelines for submitting code & software for further information.

## Data

Policy information about availability of data

All manuscripts must include a data availability statement. This statement should provide the following information, where applicable:
- Accession codes, unique identifiers, or web links for publicly available datasets
- A description of any restrictions on data availability
- For clinical datasets or third party data, please ensure that the statement adheres to our policy

All data generated for this study, including measurements and data analysis results are available at https://osf.io/wunfd/

# Research involving human participants, their data, or biological material

Policy information about studies with human participants or human data. See also policy information about sex, gender (identity/presentation), and sexual orientation and race, ethnicity and racism.

| | |
|---|---|
| Reporting on sex and gender | N/A |
| Reporting on race, ethnicity, or other socially relevant groupings | N/A |
| Population characteristics | N/A |
| Recruitment | N/A |
| Ethics oversight | N/A |

Note that full information on the approval of the study protocol must also be provided in the manuscript.

# Field-specific reporting

Please select the one below that is the best fit for your research. If you are not sure, read the appropriate sections before making your selection.

☐ Life sciences ☐ Behavioural & social sciences ☒ Ecological, evolutionary & environmental sciences

For a reference copy of the document with all sections, see nature.com/documents/nr-reporting-summary-flat.pdf

# Ecological, evolutionary & environmental sciences study design

All studies must disclose on these points even when the disclosure is negative.

| | |
|---|---|
| Study description | New oxygen, carbon, nitrogen, zinc and strontium stable isotope data and radiocarbon dates of 16 sequentially sampled equid teeth; zinc and strontium stable isotope data of 24 teeth from various omnivore, herbivore and carnivore taxa to reconstruct climate and environments faced by H. sapiens groups during the Middle to Upper Palaeolithic transition at Ilsenhöhle in Ranis, Germany. |
| Research sample | Equid teeth were targeted for serial sampling as they have high-crowned teeth and are obligate drinkers that reflect oxygen isotopes of meteoric water. Additional teeth for a variety of carnivore, omnivore, and herbivore taxa were chosen for Zn and Sr analysis to explore feeding ecology across large mammals in the food web. Teeth were chosen to cover the lower part of the stratigraphic sequence, representing the MP/UP transition. |
| Sampling strategy | For equids, a sample size of >4 teeth per archaeological unit was used to guide sampling, as this has been shown to yield sufficiently precise palaeotemperature estimates (uncertainty of ~ 2-4 °C) from oxygen stable isotope measurements in European Palaeolithic palimpsest contexts (see Pryor et al., 2014 Palaeo3). Teeth from other taxa were chosen from a single archaeological unit, Layer IX, as this layer offers that largest faunal collection in the lower stratigraphic sequence. Sample sizes were constrained by availability, with an aim of 4-5 teeth per taxon and > 8 teeth per dietary group (carnivore, omnivore, herbivore). |
| Data collection | S. Pederzani collected equid tooth samples and conducted serial sampling, sample preparation and IRMS measurements for oxygen stable isotope analysis. H. Fewlass and S. Talamo conducted collagen extraction, carbon and nitrogen stable isotope analysis and radiocarbon dating on equid dentine and mandible bone samples. M. Trost collected tooth enamel samples from non-equid taxa and conducted sample preparation for zinc and strontium stable isotope analysis. N. Bourgon and J. McCormack conducted zinc and strontium isotope measurements. |
| Timing and spatial scale | Two equid tooth samples from the 2016-2022 excavation were recovered in 2019 and obtained in 2020 from the Thüringer Landesamt für Denkmalpflege und Archäologie, Weimar, Germany. No other suitable equid teeth were recovered from the 2021 or 2022 campaigns of these renewed excavations. All other tooth samples were obtained in 2018-2019 from the collection of the 1932-1938 excavation campaign housed at the Museum für Vorgeschichte, Halle (Saale), Germany. Teeth originate from a range of squares across the extent of the excavations and square and depth information is given in Supplementary Table 1. |
| Data exclusions | In few cases, individual oxygen stable isotope measurements (of triplicate analyses conducted for each sample) were excluded if predetermined IRMS quality control criteria of peak shape and the relationship of sample amount to peak area did not conform to good quality measurements. In these cases, oxygen stable isotope delta values represent the average of two, rather than the typical three measurements per sample. |
| Reproducibility | All stable isotope analyses (O, C, N, Zn, Sr) were repeated on at least a subset of samples (all samples in triplicate for oxygen, all samples in duplicate for carbon and nitrogen, a subset in duplicate for Zn and Sr isotope analysis) to determine analytical reproducibility. Details of analytical reproducibility are described in Supplementary Text 5 (Extended methods). Additionally, all code and data to reproduce the manuscript text, figures, tables, statistical analyses, inverse modelling and temperature estimation are supplied in an associated online repository. |

| Randomization | N/A |
|---|---|
| Blinding | N/A |

Did the study involve field work? ☒ Yes ☐ No

## Field work, collection and transport

| Field conditions | Excavations at Ilsenhöhle in Ranis were conducted from 2016-2022. Two equid teeth used in this study were recovered in July/ August 2019. |
|---|---|
| Location | All specimens were recovered from Ilsenhöhle in Ranis, Germany (50°39.7563'N, 11°33.9139'E). |
| Access & import/export | Samples were obtained from the Thüringer Landesamt für Denkmalpflege und Archäologie (TLDA), Weimar, Germany and the Landesamt für Denkmalpflege und Archäologie Sachsen-Anhalt, Museum für Vorgeschichte (LDA), Halle (Saale), Germany. Sampling was conducted at the MPI-EVA, Leipzig, Germany without need for exporting. Permissions for destructive sampling were given by the LDA on 18.04.2018 (Nr. 14/2018) and by the TLDA on 26.11.2019 (Vorgangsnummer 16/116). |
| Disturbance | The samples were obtained from excavations of the archaeological site. The area of the renewed excavations was kept as small as possible to reach the lowest layers following safety measures of stepped excavation levels. |

# Reporting for specific materials, systems and methods

We require information from authors about some types of materials, experimental systems and methods used in many studies. Here, indicate whether each material, system or method listed is relevant to your study. If you are not sure if a list item applies to your research, read the appropriate section before selecting a response.

### Materials & experimental systems

| n/a | Involved in the study |
|---|---|
| ☒ | ☐ Antibodies |
| ☒ | ☐ Eukaryotic cell lines |
| ☐ | ☒ Palaeontology and archaeology |
| ☒ | ☐ Animals and other organisms |
| ☒ | ☐ Clinical data |
| ☒ | ☐ Dual use research of concern |
| ☒ | ☐ Plants |

### Methods

| n/a | Involved in the study |
|---|---|
| ☒ | ☐ ChIP-seq |
| ☒ | ☐ Flow cytometry |
| ☒ | ☐ MRI-based neuroimaging |

## Palaeontology and Archaeology

| Specimen provenance | Samples were obtained from the Thüringer Landesamt für Denkmalpflege und Archäologie (TLDA), Weimar, Germany and the Landesamt für Denkmalpflege und Archäologie Sachsen-Anhalt, Museum für Vorgeschichte (LDA), Halle (Saale), Germany. Sampling was conducted at the MPI-EVA, Leipzig, Germany without need for exporting. Permissions for destructive sampling were given by the LDA on 18.04.2018 (Nr. 14/2018) and by the TLDA on 26.11.2019 (Vorgangsnummer 16/116). Specimen IDs issued by the museums are reported for all specimens in Supplementary Table 1 |
|---|---|
| Specimen deposition | All specimens have been returned to the LDA and the TLDA, where they are curated under museum authority. |
| Dating methods | 16 equid samples collected for this study were pretreated and measured for 14C dating as part of this study. Collagen extraction and purification (including ultrafiltration) was carried out at the MPI-EVA, Leipzig using published protocols, which are described in the methods section and the Supplementary Extended methods. The suitability of collagen extracts for measurement was assessed based on coll % yield, elemental data (C%, N%, C:N). Quality criteria for all samples is included in the Supplementary Extended methods and in Supplementary Table 2. Samples were graphitised and measured with a MICADAS AMS at ETH-ZURICH and MAMS. Both uncalibrated and calibrated dates and laboratory codes are reported in Supplementary Table 2. Dates were calibrated using OxCal 4.3 using the IntCal20 data set. |

☒ Tick this box to confirm that the raw and calibrated dates are available in the paper or in Supplementary Information.

| Ethics oversight | Permissions for destructive sampling were given by the LDA by the TLDA, who are the relevant archaeological authorities regulating protection of archaeological finds in Thuringia and Saxony-Anhalt, Germany. |
|---|---|

Note that full information on the approval of the study protocol must also be provided in the manuscript.

