## [Peer Review File · Nature Ecology & Evolution]

Peer Review Information

Journal: Nature Ecology & Evolution

Manuscript Title: Early Homo sapiens dispersed into cold-arid steppes in central Europe

Corresponding author name(s): Sarah Pederzani

Editorial Notes:

Reviewer Comments & Decisions:

Decision Letter, initial version:

1st September 2023

Dear Sarah,

Your Article, "Early Homo sapiens dispersed into cold-arid steppes in central Europe" has now been seen by three reviewers. You will see from their comments copied below that while they find your work of considerable potential interest, they have raised quite substantial concerns that must be addressed. In light of these comments, we cannot accept the manuscript for publication, but would be very interested in considering a revised version that addresses these serious concerns.

We hope you will find the reviewers' comments useful as you decide how to proceed. If you wish to submit a substantially revised manuscript, please bear in mind that we will be reluctant to approach the reviewers again in the absence of major revisions.

In particular, reviewers 3 and 4 identify some technical issues with the sample and interpretation thereof that we will need to see addressed in order for us to consider a future revision.

If you choose to revise your manuscript taking into account all reviewer and editor comments, please highlight all changes in the manuscript text file [OPTIONAL: in Microsoft Word format].

* Include a "Response to reviewers" document detailing, point-by-point, how you addressed each referee comment. If no action was taken to address a point, you must provide a compelling argument. This response will be sent back to the referees along with the revised manuscript.

* If you have not done so already we suggest that you begin to revise your manuscript so that it conforms to our Article format instructions at <http://www.nature.com/natecolevol/info/final-submission>. Refer also to any guidelines provided in this letter.

2* Include a revised version of any required reporting checklist. It will be available to referees (and, potentially, statisticians) to aid in their evaluation if the manuscript goes back for peer review. A revised checklist is essential for re-review of the paper.

[REDACTED]

If you wish to submit a suitably revised manuscript we would hope to receive it within 6 months. If you cannot send it within this time, please let us know. We will be happy to consider your revision so long as nothing similar has been accepted for publication at Nature Ecology & Evolution or published elsewhere.

Nature Ecology & Evolution is committed to improving transparency in authorship. As part of our efforts in this direction, we are now requesting that all authors identified as 'corresponding author' on published papers create and link their Open Researcher and Contributor Identifier (ORCID) with their account on the Manuscript Tracking System (MTS), prior to acceptance. This applies to primary research papers only. ORCID helps the scientific community achieve unambiguous attribution of all scholarly contributions. You can create and link your ORCID from the home page of the MTS by clicking on 'Modify my Springer Nature account'. For more information please visit www.springernature.com/orcid.

Thank you for the opportunity to review your work.

[REDACTED]

Reviewer expertise:

Reviewer #1: Middle/Upper Palaeolithic archaeology (note that this reviewer has reviewed all three papers submitted to Nature and Nature Ecology & Evolution)

Reviewer #2: human environment interactions

Reviewer #3: palaeoenvironmental reconstructions, isotopes (has also reviewed NATECOLEVOL-23061426)

Reviewer #4: signed report

2Reviewers' comments:

Reviewer #1 (Remarks to the Author):

This article examines stable isotopes measured on horse teeth recovered from Ilsenhöhle in an effort to characterize the climatic and environmental conditions present in the region during the site's LRJ occupations, as well the immediately younger levels.

A caveat upfront – I'm not a stable isotope specialist so will leave the detailed review of those aspects of the paper to my specialist colleagues. That said, I find the presentation of the data, methods, and results to be clear.

Following are some general comments that can hopefully serve to improve the paper in the event that it is accepted, which to me would seem a reasonable decision.

First, I would eliminate the reference to Mandrin (Slimak et al. 2022). However, if the authors wish to keep it, then its inclusion warrants being heavily qualified. The reasons being that many colleagues who specialize in this portion of the Paleolithic remain unconvinced of the claims made by Slimak and colleagues. They make their claim of a *H. sapiens* present in Eastern France at 54 ka based on the presence of a single modern tooth in a Neronian level. However, all archaeologists are aware of the potential stratigraphic mobility of a single tooth, yet no refitting analyses or related data were presented with the publication, and the tooth was recovered in a portion of the site where stratigraphic mobility is highly probable. So, while it's true that population dynamics during the initial UP are a complex patchwork of multiple dispersals and regional cultural trajectories, I think that for the moment, Mandrin does not warrant being cited in that mix.

On line 78, I would not use the term "niche". The reason being that the data obtained from one site do not provide us a vision of the niche that was exploited by human populations, neither from an Grinnellian or an Eltonian standpoint. Furthermore, without explaining whether one is specifically referring to a realized niche or an existing fundamental niche, again, the data from one site do not allow one to estimate either.

Line 187 – I do not think that "extensive chronological overlap" is the way to express this. Two of the overlaps are at the extreme ends of the error bars so we're really only dealing with half of the samples that overlap. Unless I am missing something? In fact, it would seem that the site was used sporadically across a range of climatic conditions, some of which occurred during full stadials.

This subject comes up again in line 221, but there is a caveat given in lines 223–224 that effectively negates an unequivocal and predominant assignment to GS12. In my review of the parent article submitted to Nature, my complementary chronological analysis (shared with those authors and that I also include here as an attachment) serves to extend a bit the lower range of level 9 and also corroborates the OxCal analyses showing that the posterior intervals cover H5 (GS13), GI12 and GS12. I get the impression that the articles are trying to place the LRJ primarily in GS12, but I see occupations as being possible in a range of stadial conditions and the intervening interstadial. Finally, I'm confused by the succession of GI and GS provided in line 227. I would reverse the order from

3oldest to youngest, but maybe that's what the authors did but instead meant to say GI12-GS12-GI11. Is this the case? In any event, I did not follow what currently written.

Finally, on line 356, I would replace "may represent" with "likely represent".

Overall, a clear, well-written article with interesting and pertinent results.

Reviewer #2 (Remarks to the Author):

This work deals with geochemical evidence for early Homo sapiens dispersal throughout the very cold and arid steppes of central Europe. First author of the manuscript is S. Pederzani, the last one being J.-J. Hublin.

The main conclusion of this work is given at the end of the abstract with the following words « ... this demonstrates that humans (H. sapiens) operated (appropriate word ??) in severe cold conditions during multiple distinct early dispersals into Europe and suggests pronounced adaptability. »

I do not think that manuscript is a demonstration that H. sapiens was especially adapted to severe cold and arid conditions during early dispersal in Europe. Aridity is not proven in this study as I explained below and temperatures were not so low. Considering the calculated range (mean $\delta^{18}O_w$ comprised between -16 and -13‰) of water drunk by horses – and assuming that it reflects unmodified meteoric waters – climate about 45,000 years ago in Germany ($\approx 51^\circ N$) would have resembled those prevailing today at about $50^\circ N$ in North America or close to $60^\circ N$ in northern Europe and Asia (northern Finland and Norway, northwestern Siberia). These areas are not considered today as very cold and arid. Such discussion is missing.

Moreover, if the authors defend the hypothesis of prevailing very cold conditions and aridity, it remains unusual living environments for horses (that did not migrate on the basis of Sr isotopes?) even if it cannot be totally excluded. What about the presence of reindeer on the studied area for example? This point must be discussed.

- I must recognize that this manuscript contains interesting data, especially oxygen isotopes from apatite phosphate and ^{14}C data. However, the multi-isotopic proxy approach is not convincing as the Zn, C and Sr isotopic data are not very useful here, except to open doors that are already open (line 234: "Zinc stable isotope values of the food web at Ranis follow expected trophic level relationships with low values in carnivores and high values in herbivores"). It is especially true when the alteration of apatite or collagen are not evaluated and discussed.
- $\delta^{15}N$ data are misinterpreted because it is known that horse collagen is commonly characterized by anomalous ^{15}N -enrichment relative to herbivorous vertebrates from the same trophic level. It means here that the measured high $\delta^{15}N$ values are not evidence of aridity in the living environment of horses, independent arguments should be provided. See for example the work published by Van Klinken et al. (2002), Sponheimer et al. (2003) and Kuitens et al. (2015):

- Kuitens M, van Kolfschoten T, van der Plicht J (2015) Elevated $\delta^{15}\text{N}$ values in mammoths: a comparison with modern elephants. *Archaeol Anthropol Sci* 7(3):289–295. <https://doi.org/10.1007/s12520-012-0095-2>
 - Sponheimer M, Robinson T, Ayliffe L, Roeder B, Hammer J, Passey B, West A, Cerling T, Dearing D, Ehleringer J (2003) Nitrogen isotopes in mammalian herbivores: hair $\delta^{15}\text{N}$ values from a controlled feeding study. *Int J Osteoarchaeol* 13(1–2):80–87. <https://doi.org/10.1002/oa.655>
 - Van Klinken GJ, Richards MP, Hedges BEM (2002) An overview of causes for stable isotopic variations in past European human populations: environmental, ecophysiological, and cultural effects. In: Ambrose SH, Katzenberg MA (eds) *Biogeochemical approaches to paleodietary analysis*. Springer, US, Boston, MA, pp 39–63
- Temperature reconstructions are only based on horse remains, it should have been done for several taxa to check whether or not the horse isotopic record has been biased by its ecology, diet, ethology ...
- Air temperatures calculations lack some critical information: 1) what oxygen isotope fractionation equation has been used knowing that discrepancies were reported in the literature, 2) what about the "dampening effect" (Passey and Cerling, 2002) that minimizes the real seasonal temperature variations? 3) what equation has been used to relate the $\delta^{18}\text{O}$ to air temperature values? 4) how errors have been propagated through the various measurements and equations to obtain the temperature uncertainties?

All this information should appear in the main text of the publication

PASSEY, B.H. AND CERLING, T.E., 2002, Tooth enamel mineralization in ungulates: implications for recovering a primary isotopic time-series: *Geochimica et Cosmochimica Acta*, v. 66, p. 3225–3234.

- The supplementary material file is huge and one should expect that all information is available, especially concerning the analytical methods, however, I had the displeasure of reading that a home-made isotopic standard for oxygen isotope analysis of phosphates was produced without taking into account the method recently published by Lécuyer et al. (2019) providing the required experimental conditions (kinetics of isotopic exchange, isotopic fraction between dissolved phosphate and water.

Line 566 of supplementary material: "... an in-house silver phosphate standard (KDHP.N, $\delta^{18}\text{O} = 4.2 \pm 0.3 \text{‰}$, 1s.d.). This in-house standard was obtained by equilibrating a KH_2PO_4 solution made with Leipzig winter precipitation at ca. 140 °C for several days ..."

Missing reference:

Lécuyer, C., Fourel, F., Seris, M., Amiot, R., Goedert, J., & Simon, L. (2019). Synthesis of In-House Produced Calibrated Silver Phosphate with a Large Range of Oxygen Isotope Compositions. *Geostandards and Geoanalytical Research*, 43(4), 681–688.

My opinion is that this work needs major revisions before it can be published with a clear focus on the rigorous reconstruction of air paleotemperatures on the basis of oxygen isotope data with a discussion of the limitations associated with the use of fossil horse dental remains for this period.

5Sincerely yours

Prof. Dr. Christophe Lécuyer

Reviewer #3 (Remarks to the Author):

Overall comments:

Congratulations to all the authors for this impressive and exciting piece of work. The paper presents a robust methodology and significant results relevant for understanding the environment of the earliest incursion of Homo sapiens in central Europe. My review consists of minor suggestions for changes and clarification as elaborated below.

Main text – wording and clarifications:

Line 45: I suggest rewording 'multi-stable isotope record' in the abstract to e.g. 'multiple stable isotope records' to i) align with the way you describe this in the main text and ii) because multi-stable isotope' with the hyphen is linguistically unclear.

Line 77: I recommend changing the wording: 'data generated here' to something more specific/in line with the argument, e.g. 'isotopic data generated here'/'climate proxy generated here'.

Line 78: wording: 'initial rapid spread' is an interpretation of data, which is not presented/demonstrated in the paragraph/at this stage in manuscript. I recommend changing to 'initial spread' or adding information that demonstrates/refers/hints to data that shows that the initial spread was rapid (what is rapid in your definition?) – or e.g. by citing the accompanying Ranis chronology paper as you do later in the manuscript where you discuss the rapid spread of the LRJ.

Line 130-1031: Starting the sentence with 'otherwise' is not easily understandable (negating what?): "Otherwise, the correlation between $87\text{Sr}/86\text{Sr}$ and $\delta 66\text{Zn}$ seems driven by two outliers and has a very shallow slope, suggesting that $\delta 66\text{Zn}$ values are predominantly driven by diet". I suggest removing 'otherwise' or editing the sentence – e.g. "The correlation between $87\text{Sr}/86\text{Sr}$ and $\delta 66\text{Zn}$ observed in Supplementary Figure 8 may be driven by two outliers (hyena) and has a very shallow slope, suggesting that $\delta 66\text{Zn}$ values are predominantly driven by diet".

Line 200: It would be useful if you added the low $\delta 18\text{O}$ phos range from Ranis in brackets at the end of this sentence to help the reader: 'Equid $\delta 18\text{O}$ phos data reported here are among Europe's lowest ever reported for MIS 5 to MIS 3 ($\sim x-x \text{‰}$)'.

Line 207: As the previous comment, I suggest adding the absolute comparative information in this sentence: 'The data from $\sim 45\text{--}20743$ ka cal BP falls $\sim 0.5\text{--}1.5 \text{‰}$ lower than even the lowest equid $\delta 18\text{O}$ phos data reported ($= X \text{‰}$) from the Initial Upper Palaeolithic (IUP) occupation of Bacho Kiro Cave, Bulgaria, which has been used to reconstruct subarctic climatic conditions for an early presence of H. sapiens in south-eastern Europe8.'

Line 2014-2016: Wording - It is unclear to me what you mean with 'departures' in the following sentence: 'Palaeotemperature estimates show mean annual temperatures below freezing for the 215 coldest interval at $\sim 45\text{--}43$ ka cal BP, and temperatures $\sim 7\text{--}15 \text{°C}$ below modern-day temperatures with strongest departures in winter'. Departures in what direction and from what baseline? Can you find more clear language to describe the function?

6Line 232-233 & lines 240-242: I recommend adding the values in brackets at the end of the sentence to help the reader and to clarify what you define as 'remarkably high' in the following sentence: 'Remarkably high $\delta^{15}\text{N}$ values of equids in the ~45–43 ka cal BP interval suggest either a hypergrazer feeding ecology or dry soil conditions (~X-X $\delta^{15}\text{N}$)'. The same goes for $\delta^{13}\text{C}$ in lines 240-242.

Methods: no comments.

Figures:

Figure 2: If it can be done aesthetically, it would be helpful if the modern comparative data (lines and shaded ribbons) in the figure was labeled in the actual figure space, not just in the text. E.g. by adding a small text saying 'modern reference'. This also applies to the Extended data figure 3.

Figure 3:

A) The purple thick lines used to indicate the 95% probability layer age is a little difficult to immediately understand when viewing the figure. I suggest maybe thickening the lines even more and making the colour more shaded or in another way visually differentiate them more from the direct hominin samples.

B) It appears that the hominin samples from lower layers (~XI) date younger than the hominin samples from the top layers (~IX) from the Hülle collection. This is opposite than you would expect and I am missing an explicit discussion/mention of this in the text. The overall problems with the old collection and layer attribution is adequately discussed in the paper, but the almost inverse chronology is quite striking in this figure.

Extended data:

Extended data figure 1: It is difficult to see the written Ranis in the top left map because it is written in white on a white background map. I suggest changing to text colour to black.

Supplementary text: no comments.

Supplementary Figures:

Supplementary Figure 2: in the figure text, I recommend adding colour specification in brackets: "Colour-marked summer (red) and winter (blue) season measurements were extracted"

Supplementary Tables: No comments.

Links to data repositories are checked and functional.

References: no comments

Signed: Trine Kellberg Nielsen

Reviewer #4 (Remarks to the Author):

This paper presents the results of multi stable isotope analysis of faunal remains in order to undertake palaeotemperature, and palaeoenvironmental reconstruction from the site of Ilsenhöhle Ranis, and is part of a suite of submissions related to new findings at this site. This paper uses $\delta^{18}\text{O}$ analysis to reconstruct palaeotemperatures the site in addition to $\delta^{13}\text{C}$ and $\delta^{15}\text{N}$ values from tooth dentine collagen (and two bone collagen samples) in addition to $^{87}\text{Sr}/^{86}\text{Sr}$ and $\delta^{66}\text{Zn}$ analysis to make the argument that Anatomically Modern humans dispersed into cold, arid steppe environments of Central Europe. The findings provide a valuable insight into what environmental and climatic conditions were like when Modern Humans were occupying this part of the world.

7The research combines results from the old von Breitenbuch and Hülle excavations, in addition to the more recent excavations undertaken by the MPI-EVA. The challenges of using the archive collection were discussed in the main article and supplementary information, particularly regarding relating the boxes of specimens with sediment descriptions. The Equid teeth used were directly dated, meaning that the chronology of the specimens analysed has been established. There is still need for caution about the connection of the specimens to anthropogenic activity, the authors show that there is an overlap between the chronologies of the specimens analysed, and periods when the site was occupied by humans. However, given that fluctuations in temperature on a millennial scale has been observed in other proxies, the palimpsestic nature of the archaeological record, the challenges of using archive collections (e.g. basing level attributions on soil colour descriptions listed on boxes, as outlined in SFig1), and the fact that radiocarbon errors for this time can be relatively large, the assertion that the specimens analysed do relate to periods of human activity is not assured. The related manuscript outlining the results of the zooarchaeological analysis demonstrate that carnivore activity in some of the levels (e.g. 7) was high, indicating that specimens may represent periods of carnivore activity, rather than being accumulated by human agents. In line 272-273 the authors mention that evidence for anthropogenic modifications were low, and lines 266-266 state that human occupation at the site was ephemeral, representing either short term occupation or site-specific tasks. The related manuscript on the zooarchaeological remains states that "between 55,000 and 40,000 years ago (Layers 12-7) "the large cave Ilsenhöhle at Ranis was predominantly used for hyaena denning and cave bear hibernation" (Lines 357-358), indicating that anthropogenic activity was low at the site. Based on this, it is not entirely convincing that the stable isotope results can necessarily be linked to periods of human activity. It would be beneficial for the authors to show a greater awareness of these limitations in linking the stable isotope results to periods when humans were actually occupying the site, and to strengthen the wider archaeological argument linking the specimens sampled to anthropogenic activity. If this isn't possible then a more conservative manuscript title may be more appropriate.

The sample size of 16 equid teeth is large for this kind of study, and the findings of the $\delta^{18}\text{O}$ phos have allowed for temperature reconstructions that have yielded some intriguing results, particularly regarding the lowest $\delta^{18}\text{O}$ phos values for MIS5 to MIS3 which is notable. The methods and approach used are well justified and scientifically sound, and the interpretations of the stable isotope data are well explained, justified and supported with wider evidence. Key literature regarding isotope systems is cited in the manuscript in the supplementary information files, which clearly explain the processes influencing the values observed. The arguments, thought processes and justifications for the interpretations of the stable isotope results given in the manuscript are comprehensively addressed in the supplementary information files. The application of Zinc here was novel and adds valuable data to our understanding of this isotope in relation to dietary ecology for this period.

The supplementary files are comprehensive, with raw data provided, but many of these are essential to having a full understanding of the site and material, and interpretations relied on. I suspect word count limitations are an issue here preventing some of these important details from being included in the main manuscript.

An area to address in the main manuscript is clarity about which tissues were analysed for the isotopes being discussed, further signposting in the manuscript, and figures would be helpful. Some of the areas where this needs attention has been detailed in the specific comments below.

Figure 3- It would be worth considering the addition of the GRIP and/or NGRIP record results here to show how they relate to the stable isotope results achieved from the samples.

57- Homo neanderthalensis should be used here initially, the term Neanderthals will be sufficient subsequently.

148- Please clarify in the text which tissue the $\delta^{13}\text{C}$ values are from.

232- please clarify in the text which tissue the $\delta^{15}\text{N}$ values were measured in.

238-239- Are there other environmental proxies that can also be referred to support the hypothesis of an open steppe environment?

326- Please refer to the formal notation of isotope analysis ($\delta^{18}\text{O}_{\text{Phos}}$, $\delta^{15}\text{N}$ etc), not just C, O, N etc here.

323-324- The methodology here states that for some samples mandibular bone collagen was used instead of tooth dentine. It wasn't immediately clear in the results and figures which values were referring to. Please clarify this in the text.

Line 327- 'Tooth enamel specimens' would be a better term to use here.

Author Rebuttal to Initial comments

Response to Reviewers

We would like to thank the reviewers for their in-depth engagement with our paper and the supplementary data and for their thoughtful and constructive comments. We have implemented the vast majority of changes requested by the reviewers, which has greatly improved the manuscript. Some important changes include:

- a more clear discussion of the climatic variability seen in the stable isotope record and the implications regarding correlations with climatic events
- additional data and extensive rewrites to better support the relevance of our climatic data with human presence at the site. This includes substantial revisions to explicitly discuss the limitations incurred by analysing fauna accumulated mostly by carnivores.
- added justification of the multi-proxy stable isotope approach and an improved description of the specific climatic conditions reconstructed with reference to closest modern analogues
- justification of why only one taxon was used with a discussion of caveats of this approach

9- additional signposts and descriptions of how to obtain the methodological information regarding the oxygen isotope inverse model and the equations used for palaeotemperature estimation

We detail below the original comments from the reviewers and responses by the authors (in blue) to each point.

Please note that electronic supplementary material such as data and analysis code can be accessed at https://osf.io/vjhy7/?view_only=2d28f8e9c0c1498faaddb5c6d0d1b69e.

Reviewer #1

This article examines stable isotopes measured on horse teeth recovered from Ilsenhoehle in an effort to characterize the climatic and environmental conditions present in the region during the site's LRJ occupations, as well the immediately younger levels.

A caveat upfront – I'm not a stable isotope specialist so will leave the detailed review of those aspects of the paper to my specialist colleagues. That said, I find the presentation of the data, methods, and results to be clear.

Following are some general comments that can hopefully serve to improve the paper in the event that it is accepted, which to me would seem a reasonable decision.

First, I would eliminate the reference to Mandrin (Slimak et al. 2022). However, if the authors wish to keep it, then its inclusion warrants being heavily qualified. The reasons being that many colleagues who specialize in this portion of the Paleolithic remain unconvinced of the claims made by Slimak and colleagues. They make their claim of a *H. sapiens* present in Eastern France at 54 ka based on the presence of a single modern tooth in a Neronian level. However, all archaeologists are aware of the potential stratigraphic mobility of a single tooth, yet no refitting analyses or related data were presented with the publication, and the tooth was recovered in a portion of the site where stratigraphic mobility is highly probable. So, while it's true that population dynamics during the initial UP are a complex patchwork of multiple dispersals and regional cultural trajectories, I think that for the moment, Mandrin does not warrant being cited in that mix.

We appreciate the reviewer's caution regarding the inclusion of the reference to the Grotte Mandrin potential *H. sapiens* molar, a sentiment that is not uncommon in the field. However, we do feel that simply excluding the study would misrepresent the current state of the debate in

10this field, as the Mandrin study has, despite associated uncertainties, formed an important part of the recent discussion around *H. sapiens* dispersal into Europe. We therefore wish to keep this reference at this point in the manuscript. On the other hand, we feel that an extensive debate of that study's merits or lack thereof is beyond the scope of our paper and would detract from the focus of the introduction. To achieve a compromise between these considerations we have now included a reference to a discussion article which quotes the criticisms of multiple colleagues regarding the Mandrin study.

On line 78, I would not use the term "niche". The reason being that the data obtained from one site do not provide us a vision of the niche that was exploited by human populations, neither from an Grinnellian or an Eltonian standpoint. Furthermore, without explaining whether one is specifically referring to a realized niche or an existing fundamental niche, again, the data from one site do not allow one to estimate either.

We have now rephrased this to "*climatic and environmental conditions that pioneering H. sapiens groups exploited*".

Line 187 – I do not think that "extensive chronological overlap" is the way to express this. Two of the overlaps are at the extreme ends of the error bars so we're really only dealing with half of the samples that overlap. Unless I am missing something? In fact, it would seem that the site was used sporadically across a range of climatic conditions, some of which occurred during full stadials.

We have now rephrased this to just say "[...] confirms that the chronological overlap between *H. sapiens* specimens and the coldest phase at ~45-43 ka cal BP is independent of the calibration curve [...]". In response to a similar comment by another reviewer we have now also added more discussion on the relationship between our climatic data and human presence at the site in the results, discussion and methods section. One of the changes also includes adding a comparison not only with the directly dated *H. sapiens* remains but also with human-modified faunal fragments, which we have now also included in Figure 3 and Supplementary Figure 13.

We address the overlap of *H. sapiens* presence with a range of climatic conditions in our response to the following comment.

This subject comes up again in line 221, but there is a caveat given in lines 223–224 that effectively negates an unequivocal and predominant assignment to GS12. In my review of the parent article submitted to Nature, my complementary chronological analysis (shared with those authors and that I also include here as an attachment) serves to extend a bit the lower

range of level 9 and also corroborates the OxCal analyses showing that the posterior intervals cover H5 (GS13), GI12 and GS12. I get the impression that the articles are trying to place the LRJ primarily in GS12, but I see occupations as being possible in a range of stadial conditions and the intervening interstadial.

We agree with the reviewer's point that comparisons of the radiocarbon dates of the site with long-term climatic records of millennial-scale climatic change to assign specific stadial/interstadial events are challenging and we acknowledge that generally the chronology of the LRJ overlaps with a number of Greenland stadials and interstadials (e.g. GS13-GI12-GS12). Indeed, we did not intend to suggest that the LRJ occupation exclusively coincides with GS12, although we appreciated that this was not as clear as we had hoped. We originally highlighted the overlap with GS12 (among other phases) in conjunction with the stable isotope evidence of low temperatures because we find this result particularly interesting in contrast with prevalent models suggesting that early forays of *H. sapiens* exclusively took place during warm climatic phases.

We have now rephrased this section of our manuscript to avoid the impression that the LRJ occupations only took place during GS 12 (see lines 232-236 and lines 246-252). However, we do think that it is pertinent to emphasise that while climatic variability congruent with a cooling from a GI into a GS can be seen in the stable isotope data, even the comparatively warmer phases are still significantly colder (MAT < 5°C) than modern-day conditions. We therefore interpret our evidence as showing climatic variability within different degrees of cold to very cold climatic conditions similar to those in the subarctic to tundra climate classifications. We have now made this more explicit in the text, as we believe that simply referring to a variety of climatic conditions or to interstadial conditions without further qualification could be misunderstood to mean temperate climatic conditions.

We have made similar changes in the parent publication submitted to Nature.

Finally, I'm confused by the succession of GI and GS provided in line 227. I would reverse the order from oldest to youngest, but maybe that's what the authors did but instead meant to say GI12-GS12-GI11. Is this the case? In any event, I did not follow what currently written.

Indeed, this should have been GI 12-GS12-GI11. We apologise for this error. This section has now been rephrased in response to an earlier comment by this reviewer and no longer includes this particular statement.

Finally, on line 356, I would replace "may represent" with "likely represent".

Changed.

Overall, a clear, well-written article with interesting and pertinent results.

Reviewer #2

This work deals with geochemical evidence for early Homo sapiens dispersal throughout the very cold and arid steppes of central Europe. First author of the manuscript is S. Pederzani, the last one being J.-J. Hublin.

The main conclusion of this work is given at the end of the abstract with the following words « ... this demonstrates that humans (*H. sapiens*) operated (appropriate word ??) in severe cold conditions during multiple distinct early dispersals into Europe and suggests pronounced adaptability. »

I do not think that manuscript is a demonstration that *H. sapiens* was especially adapted to severe cold and arid conditions during early dispersal in Europe. Aridity is not proven in this study as I explained below and temperatures were not so low.

Considering the calculated range (mean $\delta^{18}\text{O}_{\text{mw}}$ comprised between -16 and -13‰) of water drunk by horses – and assuming that it reflects unmodified meteoric waters – climate about 45,000 years ago in Germany ($\approx 51^\circ\text{N}$) would have resembled those prevailing today at about 50°N in North America or close to 60°N in northern Europe and Asia (northern Finland and Norway, northwestern Siberia). These areas are not considered today as very cold and arid. Such discussion is missing.

Using precipitation $\delta^{18}\text{O}$ data from the GNIP measurement stations, the closest analogues to our data (taking into account not only mean annual but also summer and winter values) for the coldest phase (43-45 ka cal BP) are Amderma and Pechora, which are located at ~ 65 and 69.7°N respectively and fall into polar and subarctic climate classifications with Amderma being above and Pechora close to the northern tree line. Even for the phase with slightly higher $\delta^{18}\text{O}$ (45-48 ka cal BP), $\delta^{18}\text{O}$ values most closely match the site of Rovaniemi, Finland, which is located at 66.5°N also in a subarctic climate zone.

Additionally, our temperature reconstruction for the time between 43-45 ka cal BP indicates summer temperatures below $\sim 12^\circ\text{C}$, which corresponds to the July isotherm that dictates the location of the tundra-taiga biome boundary (Callaghan et al., 2002). We are unsure why

13subarctic and polar climates with summers cold enough to prohibit tree growth in some cases would not be considered very cold.

Further, to avoid the uncertainties involved in using $\delta^{18}\text{O}_{\text{precip}}$ and temperature estimates we have also used direct comparisons with equid $\delta^{18}\text{O}$ data to support our interpretations of cold climatic conditions. In this context we feel that it is important to note, as we have done in the manuscript (line 227), that the $\delta^{18}\text{O}$ equid data from Ranis is comparable to $\delta^{18}\text{O}$ data that has been reported for the LGM (Arppe and Karhu, 2010; Kovács et al., 2012).

To avoid confusion around the types of climates we are describing specifically we have now added the specific closest analogues in terms of $\delta^{18}\text{O}_{\text{precip}}$ which can serve as illustrative examples (lines 235 and 240). Additionally, we have added that reconstructed summer conditions based on $\delta^{18}\text{O}$ and temperature estimates are unlikely to have allowed much tree growth, at least for the coldest phase (lines 241-243).

Regarding the question of aridity, we interpret the $\delta^{15}\text{N}$ and $\delta^{66}\text{Zn}$ data as being consistent with either dry soils or a pronounced grazer specialisation. This could indicate either low water availability and/or the abundance of grasses in a steppe environment. We appreciate that perhaps the possibility of both options was not clearly communicated and we have now rephrased the text in several places to reduce the emphasis on arid conditions and better communicate that our data would also be consistent with an open grassland environment. For example, in line 260 ff. it now reads:

"[...] high $\delta^{15}\text{N}$ values could indicate dry soil conditions and/or strong grazing specialisation of equids, which both imply the presence of an open steppe environment. This matches with reconstructions of grass steppe environments in central Europe during MIS 3 stadials. "

To better represent the two options, we have also made a change to the title, where we have removed the mention of aridity.

Moreover, if the authors defend the hypothesis of prevailing very cold conditions and aridity, it remains unusual living environments for horses (that did not migrate on the basis of Sr isotopes?) even if it cannot be totally excluded. What about the presence of reindeer on the studied area for example? This point must be discussed.

We are surprised by the reviewer's suggestion that cold steppe environments constitute unusual habitats for Pleistocene horses, given that they are one of the keystone species of the mammoth steppe and that equids have been most successful in colonising cold steppe environments during their evolution in the Pleistocene (Cao et al., 2023). Particularly caballoid horses are commonly considered very cold-tolerant and are one of the most common species

found in mammoth steppe assemblages including those from northern Eurasia (Cao et al., 2023; Murchie et al., 2021; Schwartz-Narbonne et al., 2019). Horse remains are common even in LGM contexts of northern Eurasian and North American localities at substantially higher latitudes than Ranis. Among many examples they are, for instance, found in LGM deposits in the southern Urals (~ 52-55 °N), southern Siberia (~54 °N), in eastern Beringia, and the Alaska North Slope (~68-70 °N) (Kosintsev and Bachura, 2013; Malikov et al., 2023; Schwartz-Narbonne et al., 2019). Furthermore, modern day semi-wild horses are found in very cold conditions and in tundra habitats including in the extreme subarctic climate of the Northern Pole of Cold (Cao et al., 2023; Zimov et al., 2012). We therefore believe that the climatic conditions reconstructed for the LRJ deposits of Ranis are fully in-line with the climatic tolerance of Late Pleistocene horses.

Regarding the presence of reindeer and also other traditionally cold-adapted fauna, we already mention (line 280) that reindeer are present in the LRJ deposits of Ranis, as are woolly mammoths, woolly rhinoceros and wolverines. In our opinion, this further supports our reconstruction of a cold climate. To further emphasise this point, we have now added that reindeer are the most abundant herbivore taxon in the LRJ deposits of the site.

- I must recognize that this manuscript contains interesting data, especially oxygen isotopes from apatite phosphate and ^{14}C data. However, the multi-isotopic proxy approach is not convincing as the Zn, C and Sr isotopic data are not very useful here, except to open doors that are already open (line 234: "Zinc stable isotope values of the food web at Ranis follow expected trophic level relationships with low values in carnivores and high values in herbivores". It is especially true when the alteration of apatite or collagen are not evaluated and discussed.

We disagree with the reviewer on the utility of multi-isotope approaches. It is ubiquitously acknowledged in the field that equifinality is a central challenge of using stable isotope approaches to reconstruct past environments and ecologies and multi-isotopic study designs are widely regarded as an important and successful tool for tackling this issue (e.g., Harris and Elliott, 2019; Jones and Britton, 2019; Lee-Thorp, 2008; Peterson and Fry, 1987; Robinson, 2022; Szpak, 2023). In particular, oxygen and nitrogen stable isotopes are notorious for being difficult to interpret without additional information from other proxies and we therefore intentionally designed our study to incorporate as many isotopic systems from the same specimens. This is also highlighted as a strength of our study by the other stable isotope reviewer (Reviewer #4). We have added this motivation of the multi-isotope approach to the beginning of the methods section (line 327 ff.).

We specifically feel that the three isotopic proxies listed by the reviewer add substantial value to our manuscript for the reasons listed below. We have now modified the text to further highlight

15why we chose to analyse these proxies, as it seems that this was not clear enough (see line 330 ff). Briefly, our reasons are as follows:

- 1) Strontium isotopic analysis is a key control to confirm absence of bias from migratory behaviour. Oxygen stable isotopes vary spatially, and climatic reconstructions can therefore be substantially impacted if animals exhibit long-distance migratory behaviour. While modern-day caballoid equids normally do not undergo long-distance migrations, the spatial ecology of large herbivores can be drastically different between the Pleistocene and Holocene, or even different Pleistocene time periods (e.g., Britton et al., 2023; Julien et al., 2012). The migratory ecology of Late Pleistocene horses is mostly unexplored and we therefore regard it as vital to provide an independent control of their movement, which we achieve using strontium isotope analysis. This control is also necessary in order to interpret the $\delta^{66}\text{Zn}$ data, as we detail in SI section 4. We have now made the importance of using $^{87}\text{Sr}/^{86}\text{Sr}$ analysis more explicit by adding additional justification to the methods section.
- 2) The application of $\delta^{66}\text{Zn}$ is still fairly new, particularly for ecological inferences beyond trophic level. This means that interpretations can be less definitive in some aspects compared to more established tracers. However, we believe that it is of particular interest here due to its relationships with plant height and soil biogeochemistry, while also adding to the literature record from a method development perspective. The reviewer seems to suggest that $\delta^{66}\text{Zn}$ is not useful in our study because it adds little new trophic level information. We agree with the reviewer that trophic level relationships in $\delta^{66}\text{Zn}$ in our sample are not surprising and match those obtained from $\delta^{15}\text{N}$ (published in the companion paper by Smith et al.). However, reconstructing trophic relationships in the food web was not at all a main goal of using $\delta^{66}\text{Zn}$ in our study, which was rather the exploration of this tracer as a tool to inform on dietary ecology of herbivores on a grazer-browser spectrum. Our intent with discussing the trophic level patterns in $\delta^{66}\text{Zn}$ was to underline the preservation of a biogenic signal, in addition to the preservation information provided by the absence of a concentration mixing line. At the same time, we wanted to show the patterns across the food web to highlight differences between herbivore taxa, such as those between equids and cervids, as this supports some of our interpretations regarding herbivore feeding ecology. It seems that the goal of using $\delta^{66}\text{Zn}$ and the implications of the patterns across the food web in our study were not conveyed clearly enough in our manuscript. We have now added a note in the methods section to clarify that our intent for the $\delta^{66}\text{Zn}$ data was to elucidate herbivore dietary ecology.

- 3) Regarding the use of $\delta^{13}\text{C}$ in our study, we feel that this proxy is important for confirming that equid diet is consistent with feeding in an open environment. As the reviewer highlights in a later comment, dietary and environmental tracers such as $\delta^{15}\text{N}$ can be complex in their influences and interpretations. The $\delta^{13}\text{C}$ data helps to constrain such data, even if we do not see pronounced diachronic changes in this proxy. Beyond this, we also feel that it would be a strange choice to exclude this particular proxy as $\delta^{13}\text{C}$ data is automatically obtained when analysing $\delta^{15}\text{N}$ of collagen. As with the other proxies, we now explain our choice of including this proxy in the beginning of the methods section.

Regarding the discussion of preservation of the different tissues, we already provide detailed information on the preservation of our collagen samples in Supplementary Text 5 (lines 689 ff.) and Supplementary Table 2 with references to this in main text lines 109 and 440. We also provide Sr and Zn concentration data and visualisations of concentration plotted against isotope data to evaluate the preservation in tooth enamel, although this aspect was only briefly discussed in the main text (now in line 479 ff.). The reliability of $\delta^{18}\text{O}$ phos and strontium isotopes in tooth enamel across geological time scales has been extensively documented in the literature, so we focus here on the less well-known $\delta^{66}\text{Zn}$. Several studies have investigated in the detail the preservation of $\delta^{66}\text{Zn}$ values, in tooth enamel and enameloid and have shown that they are well-preserved even in fossil teeth back into the Miocene (e.g., Bourgon et al., 2020; Jaouen et al., 2022; McCormack et al., 2022). We have now added more information on the diagenetic resistance of $\delta^{66}\text{Zn}$ in tooth enamel to the $\delta^{66}\text{Zn}$ background section of Supplementary Text 4 and refer to it in the main text (line 479 ff.).

- $\delta^{15}\text{N}$ data are misinterpreted because it is known that horse collagen is commonly characterized by anomalous ^{15}N -enrichment relative to herbivorous vertebrates from the same trophic level. It means here that the measured high $\delta^{15}\text{N}$ values are not evidence of aridity in the living environment of horses, independent arguments should be provided. See for example the work published by Van Klinken et al. (2002), Sponheimer et al. (2003) and Kuitems et al. (2015):

- Kuitems M, van Kolfschoten T, van der Plicht J (2015) Elevated $\delta^{15}\text{N}$ values in mammoths: a comparison with modern elephants. *Archaeol Anthropol Sci* 7(3):289–295. <https://doi.org/10.1007/s12520-012-0095-2>

- Sponheimer M, Robinson T, Ayliffe L, Roeder B, Hammer J, Passey B, West A, Cerling T, Dearing D, Ehleringer J (2003) Nitrogen isotopes in mammalian herbivores: hair $\delta^{15}\text{N}$ values from a controlled feeding study. *Int J Osteoarchaeol* 13(1–2):80–87. <https://doi.org/10.1002/oa.655>

17- Van Klinken GJ, Richards MP, Hedges BEM (2002) An overview of causes for stable isotopic variations in past European human populations: environmental, ecophysiological, and cultural effects. In: Ambrose SH, Katzenberg MA (eds) Biogeochemical approaches to paleodietary analysis. Springer, US, Boston, MA, pp 39–63

We disagree with the reviewer for two reasons.

1) While horses can sometimes exhibit higher $\delta^{15}\text{N}$ than some other herbivores, this is by no means as universal or pronounced as the reviewer suggests, as it depends on the specific dietary ecology of each herbivore species in an ecosystem. In fact, many stable isotope studies of herbivores from a variety of Pleistocene contexts, including the mammoth steppe ecosystem that we believe is most relevant for our context, show equids with similar or even lower $\delta^{15}\text{N}$ ranges than other herbivores (Bocherens et al., 1996, 2014, 2015; Britton et al., 2012, 2023; Drucker and Bocherens, 2004; Drucker et al., 2021; Naito et al., 2016; Reade et al., 2020; Richards et al., 2008; Schwartz-Narbonne et al., 2019; Stevens and Hedges, 2004; Wißing et al., 2015, 2016, 2019). In addition to these many examples, a recent overview paper by Dorothee Drucker summarising stable isotope dynamics of herbivores in the mammoth steppe effectively demonstrates that across data sets collected over the last decades equids occupy variable, including low, $\delta^{15}\text{N}$ positions in relation to other herbivores (Drucker, 2022). Even in some of the publications referenced by the reviewer horses extensively overlap in $\delta^{15}\text{N}$ with other herbivores. Controlled feeding studies even suggest that horses are more likely to exhibit lower $\delta^{15}\text{N}$ values than other herbivores as they often select low-protein forage, which leads to a lower diet-consumer spacing and lower $\delta^{15}\text{N}$ values (Sponheimer et al., 2003). The variable $\delta^{15}\text{N}$ position of horses relative to other herbivores in Pleistocene data sets as well the mechanisms elucidated in controlled feeding experiments both suggest that their $\delta^{15}\text{N}$ values are driven by changes in feeding ecology that reflects environmental change or ecological plasticity rather than being systematically determined by metabolic factors. Thus, we interpret our data accordingly.

2) Our reconstruction of dry soils or of a pronounced grazer specialisation of horses in the 43-45 ka cal BP time interval is inferred from high $\delta^{15}\text{N}$ values of horses in this time frame compared to other horse specimens in other time periods. We are therefore using a diachronic trend that is visible within this single species to make our argument. This is unrelated to the $\delta^{15}\text{N}$ relationship of horses compared to other herbivore species found at the site, and indeed we do not present $\delta^{15}\text{N}$ data from other herbivores in this study. In addition to our concerns expressed in point 1) we are, therefore, unsure how the reviewer's point invalidates our reasoning regarding the interpretation of our equid $\delta^{15}\text{N}$ data. To avoid confusion in this regard we have now rephrased the discussion of the $\delta^{15}\text{N}$ data to highlight more clearly that we are

referring to diachronic changes within the equid data, rather than comparisons with other taxa. For example, it now reads in line 253 ff.:

“Remarkably high $\delta^{15}\text{N}$ values of equid dentine and mandible bone collagen in the ~45–43 ka cal BP interval (~ 7–9 ‰) compared to earlier and later equid data suggest either a hypergrazer feeding ecology or dry soil conditions during this interval. ”

As mentioned in an earlier response we do acknowledge that the $\delta^{15}\text{N}$ interpretation as either reflecting dry soils or a grazer specialisation in an open grassland environment were not as clearly conveyed in the text as they could have been. We have now edited the text accordingly, but have made no changes suggesting that horses commonly have high $\delta^{15}\text{N}$ values as we feel like this does not accurately reflect the current picture of the published record.

- Temperature reconstructions are only based on horse remains, it should have been done for several taxa to check whether or not the horse isotopic record has been biased by its ecology, diet, ethology ...

The reviewer brings up an important point and we agree with the reviewer that using multiple taxa is preferable when estimating past temperatures. However, this requires the availability of multiple large taxa with high drinking requirements within the same deposits which is often not realistic for Palaeolithic assemblages. In our case the choice of taxon is additionally constrained as we intentionally chose to generate sub-annually resolved seasonal $\delta^{18}\text{O}$ data, for which high-crowned teeth covering at least 1 year of tooth formation are required. Such teeth additionally need to be sourced from taxa that have sufficient published information on tooth formation in order to allow the application of an inverse model to correct for seasonal amplitude damping through time-averaging. Out of the commonly available European Late Pleistocene large mammals, these requirements are only fulfilled by equids, large bovids (aurochs or bison) and mammoths, while cervids or rhinoceros lack tooth formation information, have little time recorded in single teeth, and/or don't always consume sufficiently large quantities of liquid water to be robustly tied to $\delta^{18}\text{O}$ of environmental water. Unfortunately, the Pleistocene deposits of Ranis do not contain usable numbers of either large bovid or mammoth teeth to enable a meaningful analysis of a taxon other than horses. We were therefore unable to conduct a multi-species study at this site.

That being said, we do believe that we can demonstrate that we have generated robust palaeotemperature estimates in our study. If published studies of multi-species $\delta^{18}\text{O}$ data sets from the Late Pleistocene are considered, an overview of the literature shows that results from equids commonly match well with those of a variety of other species. We have made here a figure showing some examples using data from (Fabre et al., 2011; Lécuyer et al., 2021;

19Skrzypek et al., 2011; Stephan, 2017) (see below; all values as reported in the original publications). Note that for Homos de la Peña and Laugerie-Haute E. $\delta^{18}\text{O}_{\text{ice}}$ values were used, as reported in Lécuyer et al. (2021). Rangifer $\delta^{18}\text{O}_{\text{dw}}$ values from Stephan (2017) include a humidity correction that was applied by Stephan.

It can be seen that $\delta^{18}\text{O}_{\text{dw}}$ data from equids commonly falls within the range of variation of other taxa, and also displays the same diachronic trends as those from, for example, cervids as seen in the case of Gigny. There is also no visible systematic trend by taxon in this data.

Additionally, we believe that we have made detailed arguments for why $\delta^{18}\text{O}$ of horses in our case reflects $\delta^{18}\text{O}$ of meteoric water, which are described at length in Supplementary Text 2. We also believe that the correlation of our data with climatic indicators from the sedimentological

data from the site lends further support to our climatic rather than dietary interpretation of this data.

However, while we lamentably cannot change the taxonomic composition of the available sample, we acknowledge that we could be more explicit about the fact that using multiple taxa would have been the ideal approach. We have now added a note to this effect in the methods section and discuss this aspect more extensively in Supplementary Text 2 reference to the publications mentioned above.

- Air temperatures calculations lack some critical information: 1) what oxygen isotope fractionation equation has been used knowing that discrepancies were reported in the literature, 2) what about the "dampening effect" (Passey and Cerling, 2002) that minimizes the real seasonal temperature variations? 3) what equation has been used to relate the $\delta^{18}\text{O}_w$ to air temperature values? 4) how errors have been propagated through the various measurements and equations to obtain the temperature uncertainties?

All this information should appear in the main text of the publication

PASSEY, B.H. AND CERLING, T.E., 2002, Tooth enamel mineralization in ungulates: implications for recovering a primary isotopic time-series: *Geochimica et Cosmochimica Acta*, v. 66, p. 3225–3234.

We appreciate the suggestion of the reviewer to include more methodological details in the main text rather than in the Supplementary Information. However, it appears that the reviewer has regrettably overlooked that this information is already present in extensive detail in the supplementary materials that we have provided. Specifically, we had already included a detailed description that we adjust for seasonal damping (as described in Passey and Cerling 2002) by applying the model published in Passey et al. (2005). This is mentioned in the main text methods (section entitled "Inverse modelling and palaeotemperature estimation") and extensively described in Supplementary Text 5.6. We also provide R scripts to conduct the model and all model output and parameterization information for each individual tooth. This code and data are available in the associated OSF repository, which we have linked both in the data and code availability statements and in the SI text. We have now added an additional note to the palaeotemperature section of the main text methods to make it more clear to readers how this information and data can be obtained (see line 419 ff.).

Similarly, we already include all the raw data used to establish both the $\delta^{18}\text{O}_{\text{enamel}} - \delta^{18}\text{O}_{\text{dw}}$ equation and the $\delta^{18}\text{O}_{\text{dw}}$ -air temperature equation (points 1 and 3), the R code that applies the

21equation and cite a paper where these equations are explicitly printed (main text methods line 429 ff.). The same is true for the error propagation mentioned in point 4) of the reviewer. We use the same approach already described in detail in Pryor et al. (2014) in the same implementation as described with equations in Pederzani et al. (2021), and we provide all data and R code so that readers are able to check and reproduce these calculations with minimal effort. We have now highlighted more clearly where these equations can be obtained (main text methods line 429 ff.).

Our experience with our past publications has shown that printing these equations in text form outside the code or excel sheets that apply them correctly with an appropriate error propagation often leads to their use without any error propagation. We therefore prefer to provide this information in its current format, where the necessary context of error propagation is maintained. We feel the availability of both R scripts in this publication and excel files in Pederzani et al. 2021 and Pryor et al. 2014 covers a range of formats that are accessible to a broad range of readers.

Regarding the reviewer's comment on how we justify our choice of fractionation equation to relate $\delta^{18}\text{O}_{\text{enamel}}$ and $\delta^{18}\text{O}_{\text{dw}}$ given that several equations have been published we would like to point out that we integrate all published $\delta^{18}\text{O}_{\text{enamel}}$ and $\delta^{18}\text{O}_{\text{dw}}$ data of horse known to us to derive the linear relationship between the two. We therefore do not use any of the equations published for each of the separate publications that have contributed such data. As mentioned previously, we describe this in SI section 5.7 and refer readers to this section in the main text methods (line 426 ff.). We again also provide all this collated data alongside code to reproduce the equation in the online repository, which we explicitly state in SI section 5.7. As with the other requested equations we have now added more clarification of how they can be obtained in the main text methods (line 426 ff.).

- The supplementary material file is huge and one should expect that all information is available, especially concerning the analytical methods, however, I had the displeasure of reading that a home-made isotopic standard for oxygen isotope analysis of phosphates was produced without taking into account the method recently published by Lécuyer et al. (2019) providing the required experimental conditions (kinetics of isotopic exchange, isotopic fraction between dissolved phosphate and water).

Line 566 of supplementary material: "... an in-house silver phosphate standard (KDHP.N, $\delta^{18}\text{O} = 4.2 \pm 0.3 \text{ ‰}$, 1s.d.). This in-house standard was obtained by equilibrating a KH_2PO_4 solution made with Leipzig winter precipitation at ca. 140 °C for several days ..."

Missing reference:

Lécuyer, C., Fourel, F., Seris, M., Amiot, R., Goedert, J., & Simon, L. (2019). Synthesis of In-House Produced Calibrated Silver Phosphate with a Large Range of Oxygen Isotope Compositions. *Geostandards and Geoanalytical Research*, 43(4), 681-688.

We thank the reviewer for bringing this important work to our attention. We had originally not cited this publication, as we produced our standard in late 2017 (substantially prior to the publication date of this article) based on advice of colleagues in the field. We had not updated our methodological descriptions of its production since then, and apologise for our oversight of including relevant new publications. We have now added a citation to the work mentioned by the reviewer, which serves as an excellent guide for anyone looking to repeat a similar procedure.

My opinion is that this work needs major revisions before it can be published with a clear focus on the rigorous reconstruction of air paleotemperatures on the basis of oxygen isotope data with a discussion of the limitations associated with the use of fossil horse dental remains for this period.

Sincerely yours

Prof. Dr. Christophe Lécuyer

Reviewer #3

Overall comments:

Congratulations to all the authors for this impressive and exciting piece of work. The paper presents a robust methodology and significant results relevant for understanding the environment of the earliest incursion of *Homo sapiens* in central Europe. My review consists of minor suggestions for changes and clarification as elaborated below.

Main text – wording and clarifications:

Line 45: I suggest rewording 'multi-stable isotope record' in the abstract to e.g. 'multiple stable

23isotope records' to i) align with the way you describe this in the main text and ii) because multi-stable isotope' with the hyphen is linguistically unclear.

This has been changed as suggested.

Line 77: I recommend changing the wording: 'data generated here' to something more specific/in line with the argument, e.g. 'isotopic data generated here'/'climate proxy generated here'.

This has been changed to "isotopic data".

Line 78: wording: 'initial rapid spread' is an interpretation of data, which is not presented/demonstrated in the paragraph/at this stage in manuscript. I recommend changing to 'initial spread' or adding information that demonstrates/refers/hints to data that shows that the initial spread was rapid (what is rapid in your definition?) – or e.g. by citing the accompanying Ranis chronology paper as you do later in the manuscript where you discuss the rapid spread of the LRJ.

Rephrased to "initial spread".

Line 130-1031: Starting the sentence with 'otherwise' is not easily understandable (negating what?): "Otherwise, the correlation between $87\text{Sr}/86\text{Sr}$ and $\delta 66\text{Zn}$ seems driven by two outliers and has a very shallow slope, suggesting that $\delta 66\text{Zn}$ values are predominantly driven by diet". I suggest removing 'otherwise' or editing the sentence – e.g. "The correlation between $87\text{Sr}/86\text{Sr}$ and $\delta 66\text{Zn}$ observed in Supplementary Figure 8 may be driven by two outliers (hyena) and has a very shallow slope, suggesting that $\delta 66\text{Zn}$ values are predominantly driven by diet".

We have now replaced "otherwise" with "furthermore" in this sentence.

Line 200: It would be useful if you added the low $\delta 18\text{O}$ phos range from Ranis in brackets at the end of this sentence to help the reader: 'Equid $\delta 18\text{O}$ phos data reported here are among Europe's lowest ever reported for MIS 5 to MIS 3 ($\sim x-x \text{ ‰}$).'

Changed as suggested.

Line 207: As the previous comment, I suggest adding the absolute comparative information in

this sentence: 'The data from ~45–207 43 ka cal BP falls ~0.5–1.5 ‰ lower than even the lowest equid $\delta^{18}\text{O}$ data reported (= X ‰) from the Initial Upper Palaeolithic (IUP) occupation of Bacho Kiro Cave, Bulgaria, which has been used to reconstruct subarctic climatic conditions for an early presence of *H. sapiens* in south-eastern Europe⁸.'

We have added this information as suggested.

Line 2014-2016: Wording - It is unclear to me what you mean with 'departures' in the following sentence: 'Palaeotemperature estimates show mean annual temperatures below freezing for the 215 coldest interval at ~45–43 ka cal BP, and temperatures ~7–15 °C below modern-day temperatures with strongest departures in winter'. Departures in what direction and from what baseline? Can you find more clear language to describe the function?

We have rephrased this to say "largest anomalies" rather than "departures" to clarify that we meant temperature anomalies compared to modern day conditions.

Line 232-233 & lines 240-242: I recommend adding the values in brackets at the end of the sentence to help the reader and to clarify what you define as 'remarkably high' in the following sentence: 'Remarkably high $\delta^{15}\text{N}$ values of equids in the ~45–43 ka cal BP interval suggest either a hypergrazer feeding ecology or dry soil conditions (~X-X $\delta^{15}\text{N}$)'. The same goes for $\delta^{13}\text{C}$ in lines 240-242.

Added as requested.

Methods: no comments.

Figures:

Figure 2: If it can be done aesthetically, it would be helpful if the modern comparative data (lines and shaded ribbons) in the figure was labeled in the actual figure space, not just in the text. E.g. by adding a small text saying 'modern reference'. This also applies to the Extended data figure 3.

We have added modern reference labels to the ribbons in both figures.

Figure 3:

A) The purple thick lines used to indicate the 95% probability layer age is a little difficult to immediately understand when viewing the figure. I suggest maybe thickening the lines even

more and making the colour more shaded or in another way visually differentiate them more from the direct hominin samples.

We have now made the lines darker and more opaque, have increased the spacing from the point data of the human dates and have added a “Layer age range” label into the figure to make the lines and their meaning clearer.

B) It appears that the hominin samples from lower layers (~XI) date younger than the hominin samples from the top layers (~IX) from the Hülle collection. This is opposite than you would expect and I am missing an explicit discussion/mention of this in the text. The overall problems with the old collection and layer attribution is adequately discussed in the paper, but the almost inverse chronology is quite striking in this figure.

We appreciate that the inverted-looking chronology in this figure is a bit jarring and we thank the reviewer for bringing this up. Due to the issues with the old collection the layer labels of these *H. sapiens* fragments should be taken with some caution and indeed we believe all fragments to originate from the LRJ Layer X. This is explained in more detail in the main paper submitted to Nature, and we appreciate that this should have been clearer also here. We have now pooled all *H. sapiens* dates from the old collection into a single layer group and have added a note in the figure legend to explain that while some of these were originally labelled as coming from the adjacent strata, we believe this to be caused by mixing due to rough excavation methods of the 1930s. We have adjusted Supplementary Figure 13 in the same manner.

Extended data:

Extended data figure 1: It is difficult to see the written Ranis in the top left map because it is written in white on a white background map. I suggest changing to text colour to black.

We have changed the text colour to black.

Supplementary text: no comments.

Supplementary Figures:

Supplementary Figure 2: in the figure text, I recommend adding colour specification in brackets: “Colour-marked summer (red) and winter (blue) season measurements were extracted”

We have added colour descriptions to this legend.

Supplementary Tables: No comments.
Links to data repositories are checked and functional.
References: no comments

Signed: Trine Kellberg Nielsen

Reviewer #4

This paper presents the results of multi stable isotope analysis of faunal remains in order to undertake palaeotemperature, and palaeoenvironmental reconstruction from the site of Ilsenhöhle Ranis, and is part of a suite of submissions related to new findings at this site. This paper uses $\delta^{18}\text{O}$ analysis to reconstruct palaeotemperatures the site in addition to $\delta^{13}\text{C}$ and $\delta^{15}\text{N}$ values from tooth dentine collagen (and two bone collagen samples) in addition to $^{87}\text{Sr}/^{86}\text{Sr}$ and $\delta^{66}\text{Zn}$ analysis to make the argument that Anatomically Modern humans dispersed into cold, arid steppe environments of Central Europe. The findings provide a valuable insight into what environmental and climatic conditions were like when Modern Humans were occupying this part of the world.

The research combines results from the old von Breitenbuch and Hülle excavations, in addition to the more recent excavations undertaken by the MPI-EVA. The challenges of using the archive collection were discussed in the main article and supplementary information, particularly regarding relating the boxes of specimens with sediment descriptions. The Equid teeth used were directly dated, meaning that the chronology of the specimens analysed has been established.

There is still need for caution about the connection of the specimens to anthropogenic activity, the authors show that there is an overlap between the chronologies of the specimens analysed, and periods when the site was occupied by humans. However, given that fluctuations in temperature on a millennial scale has been observed in other proxies, the palimpsestic nature of the archaeological record, the challenges of using archive collections (e.g. basing level attributions on soil colour descriptions listed on boxes, as outlined in SFig1), and the fact that radiocarbon errors for this time can be relatively large, the assertion that the specimens analysed do relate to periods of human activity is not assured. The related manuscript outlining the results of the zooarchaeological analysis demonstrate that carnivore activity in some of the

27levels (e.g. 7) was high, indicating that specimens may represent periods of carnivore activity, rather than being accumulated by human agents. In line 272-273 the authors mention that evidence for anthropogenic modifications were low, and lines 266-266 state that human occupation at the site was ephemeral, representing either short term occupation or site-specific tasks. The related manuscript on the zooarchaeological remains states that “between 55,000 and 40,000 years ago (Layers 12-7) “the large cave Ilsenhöhle at Ranis was predominantly used for hyaena denning and cave bear hibernation” (Lines 357-358), indicating that anthropogenic activity was low at the site. Based on this, it is not entirely convincing that the stable isotope results can necessarily be linked to periods of human activity. It would be beneficial for the authors to show a greater awareness of these limitations in linking the stable isotope results to periods when humans were actually occupying the site, and to strengthen the wider archaeological argument linking the specimens sampled to anthropogenic activity. If this isn't possible then a more conservative manuscript title may be more appropriate.

We agree with the reviewer that obtaining stable isotope data from anthropogenically accumulated zooarchaeological assemblages is the most robust way of connecting climatic and environmental data to human presence. The reviewer is correct that the link with human activity in this study is less direct, as indeed most faunal remains were accumulated by carnivores and there is only low intensity evidence of human activity at the site during the LRJ. The climatic data from the equid remains therefore most likely represents both periods coinciding with human activity and periods where humans were not present at the site.

To be more explicit about this limitation in establishing a direct link between stable isotope data and human activity we have now rephrased the manuscript in several places, including extensive rewrites of the results section describing the relationship of our results with the presence of *H. sapiens* at the site and the most pertinent parts of the discussion section. For example, in the discussion it now reads:

*“Our results show that climatic conditions throughout the LRJ occupations, even during the earliest phase ~48–45 ka cal BP, were characterised by temperatures substantially below modern-day conditions. Although a direct contextual connection through anthropogenic modification cannot be established, the chronological overlap between the direct dates of *H. sapiens* remains and anthropogenically-modified bone fragment with those of the equid individuals that produced the temperature results indicates that *H. sapiens* faced such subarctic to tundra climates, most likely even during this severe cold climatic phase 45–43 ka cal BP. ”*

In addition to a more explicit discussion of the limitations of our study we have added additional data to support the link with human presence as much as is possible within the characteristics of the archaeological collections available to us. We have added the following two aspects:

28- 1) We have conducted additional statistical tests comparing the radiocarbon dates of the equid remains with those of directly dated *H. sapiens* remains and have now also added the direct dates of anthropogenically-modified faunal bone fragments to our comparison to strengthen as much as we can the link with human presence at the site. This analysis (based on the χ^2 test and agreement indices of the OxCal Combine function) shows that each of the equid remains used to infer the palaeoclimate of the LRJ is statistically indistinguishable in age with one or multiple directly dated *H. sapiens* fragments and/or anthropogenically-modified faunal bone fragment, within the error range of the radiocarbon dates. This includes the two equid specimens that yielded the lowest $\delta^{18}\text{O}$ data. We now describe this in the results section of the manuscript and have added a table of these statistical comparisons to the online data repository. We attach here also a graphical representation.

- 2) While cut-marked faunal remains are few in number from the LRJ layers, these fragments do include equid remains but are dominated by reindeer. This is relevant for two reasons. 1) It is theoretically possible that carnivores and humans did not consume the same taxa as prey, but rather systematically preferred some species. The presence of cut-marked equid specimens indicates that equids were not exclusively accumulated by carnivores. Of course we are not able to extrapolate a connection to the specific equid remains samples for stable isotopes, but at least a systematic bias where equids only represent phases of carnivore activity can be excluded. 2) The dominance of reindeer in the cut-marked specimens, while based on a small number of fragments, is consistent with hunting in comparatively cold climatic conditions. The number of cut-marked specimens is too small to make strong inferences about taxon representation, but at least there is no particular indication that the taxa hunted by humans systematically differ from those present around the site during phases of carnivore activity.

We have added this information to the results section of the main text.

In the main text we have now made an effort to clarify that our data most likely also represents time periods without human occupation of the site (see quoted discussion section and the first part of the methods section). We do, however, argue that our data is broadly representative of climatic conditions that occurred during the LRJ, while substantial deviations from this are possible but not particularly likely. We agree with the reviewer that millennial-scale climatic variability during MIS 3 is significant and important to consider and our record most certainly does not capture the full climatic variability of the past, which would be impossible to achieve with any palaeoclimate proxy technique. However, we believe it to be most likely that the climatic change captured in the $\delta^{18}\text{O}$ data does capture millennial-scale climate variability of a GI-GS-GI cycle. Decadal climatic variability during the Pleistocene is documented in the North Atlantic ice core records, but is considerably less pronounced than the millennial-scale climatic change of the DO-events (Boers, 2018). We therefore argue that very strong climatic departures from the conditions that we reconstruct are relatively unlikely, especially given the lag times in vegetation recovery.

In this context it is also relevant to consider that all of the $\delta^{18}\text{O}$ data covering the lower sequence of Ranis documents cold climatic conditions with subarctic conditions or colder. Even acknowledging that we don't capture full variability and don't overlap with *H. sapiens* occupations for some time periods, it seems unlikely that none of our data would be pertinent to the human occupations of the region.

Overall, we do acknowledge that to some extent this constitutes an argument of probability and we have made an effort in the manuscript text to appropriately represent the uncertainty involved in linking our climatic data with periods of human activity at the site. Nonetheless we believe that we can document that *H. sapiens* most likely faced the climatic conditions documented here at some point of their presence in the region, which we believe is the most important aspect of documenting *H. sapiens* climate tolerance during this time period.

We have passed this sentiment down in the discussion section, where it reads as follows:

*“The ephemeral *H. sapiens* presence at the site also implies that the climatic data most likely also represents phases of site formation where humans were absent, but we argue that the temporal overlap between directly dated equids, *H. sapiens* remains and anthropogenically modified bones and the reflection of millennial-scale climatic variability in our record does suggest that we can broadly characterise the climates faced by LRJ *H. sapiens* as cold to very cold during most of the LRJ formation period. ”*

The sample size of 16 equid teeth is large for this kind of study, and the findings of the $\delta^{18}O_{phos}$ have allowed for temperature reconstructions that have yielded some intriguing results, particularly regarding the lowest $\delta^{18}O_{phos}$ values for MIS5 to MIS3 which is notable. The methods and approach used are well justified and scientifically sound, and the interpretations of the stable isotope data are well explained, justified and supported with wider evidence. Key literature regarding isotope systems is cited in the manuscript in the supplementary information files, which clearly explain the processes influencing the values observed. The arguments, thought processes and justifications for the interpretations of the stable isotope results given in the manuscript are comprehensively addressed in the supplementary information files. The application of Zinc here was novel and adds valuable data to our understanding of this isotope in relation to dietary ecology for this period. The supplementary files are comprehensive, with raw data provided, but many of these are essential to having a full understanding of the site and material, and interpretations relied on. I suspect word count limitations are an issue here preventing some of these important details from being included in the main manuscript.

Due to the large number of proxies used in the study and the word count limit of the main text we were unfortunately restricted in what we could discuss in detail in the main text rather than the SI.

An area to address in the main manuscript is clarity about which tissues were analysed for the

isotopes being discussed, further signposting in the manuscript, and figures would be helpful. Some of the areas where this needs attention has been detailed in the specific comments below.

We have now added clarifications of the tissue in the places indicated in the line-by-line comments of this reviewer as well as at the first mention of each proxy in the results and discussion section. We have also added a note to the caption of Figure 1 to state which samples were measured on collagen extracted from adhering mandible bone rather than tooth dentine.

Figure 3- It would be worth considering the addition of the GRIP and/or NGRIP record results here to show how they relate to the stable isotope results achieved from the samples.

We have now added both the NGRIP $\delta^{18}\text{O}$ record and the arboreal pollen and *Betula* pollen from the Füramoos record in western Germany to this figure to contextualise out data.

57- *Homo neanderthalensis* should be used here initially, the term Neanderthals will be sufficient subsequently.

Changed as suggested.

148- Please clarify in the text which tissue the $\delta^{13}\text{C}$ values are from.

Clarified that this refers to $\delta^{13}\text{C}$ of dentine and mandible bone collagen. We have also added a similar note in the discussion section (line 262 ff.).

232- please clarify in the text which tissue the $\delta^{15}\text{N}$ values were measured in.

We have added a clarification in this location. We have also added this clarification in the introduction (line 92) and the results section (line 144).

238-239- Are there other environmental proxies that can also be referred to support the hypothesis of an open steppe environment?

Regrettably, there is no other proxy data from the site itself that is informative on vegetation composition. Isotopic data from reindeer generally indicates lichen consumption congruent with an open environment (see companion paper by Smith et al.), but we currently do not have directly dated reindeer (or other herbivore) specimens specifically for the 43-45 ka cal BP interval, so we do not refer to it here. We are therefore restricted to comparisons with pollen records from other parts of Germany or European records with larger catchments, as we do in

33

the sentence following the section indicated by the reviewer. These records do support the expansion of grasslands and the presence of steppe environments during the Greenland stadials that could overlap with the cold phase indicated in our isotopic record (e.g., Kern et al., 2022).

326- Please refer to the formal notation of isotope analysis ($\delta^{18}\text{O}$ Phos, $\delta^{15}\text{N}$ etc), not just C, O, N etc here.

Changed as suggested.

323-324- The methodology here states that for some samples mandibular bone collagen was used instead of tooth dentine. It wasn't immediately clear in the results and figures which values were referring to. Please clarify this in the text.

We have now noted throughout the text that both dentine and mandible bone collagen were used and have clarified in the caption of Figure 1 that this concerns the two specimens from the MPI/TLDA collection. As the symbols for the different excavation collections match exactly with the dentine vs mandible bone samples we have left the shapes as is with an explanation in the caption.

Line 327- 'Tooth enamel specimens' would be a better term to use here.

Changed as suggested.

References cited in the response to reviewers

- Arppe, L.M., Karhu, J.A., 2010. Oxygen isotope values of precipitation and the thermal climate in Europe during the middle to late Weichselian ice age. *Quat. Sci. Rev.* 29, 1263–1275. <https://doi.org/10.1016/j.quascirev.2010.02.013>
- Bocherens, H., Drucker, D.G., Germonpré, M., Lázníčková-Galetová, M., Naito, Y.I., Wissing, C., Brůžek, J., Oliva, M., 2015. Reconstruction of the Gravettian food-web at Předmostí I using multi-isotopic tracking (^{13}C , ^{15}N , ^{34}S) of bone collagen. *Quat. Int., World of Gravettian Hunters* 359–360, 211–228. <https://doi.org/10.1016/j.quaint.2014.09.044>
- Bocherens, H., Drucker, D.G., Madelaine, S., 2014. Evidence for a ^{15}N positive excursion in terrestrial foodwebs at the Middle to Upper Palaeolithic transition in south-western France: Implications for early modern human palaeodiet and palaeoenvironment. *J. Hum. Evol.* 69, 31–43. <https://doi.org/10.1016/j.jhevol.2013.12.015>

- Bocherens, H., Picaud, G., Lazarev, P.A., Mariotti, A., 1996. Stable isotope abundances (^{13}C , ^{15}N) in collagen and soft tissues from Pleistocene mammals from Yakutia: Implications for the palaeobiology of the Mammoth Steppe. *Palaeogeogr. Palaeoclimatol. Palaeoecol., Biogenic Phosphates as Palaeoenvironmental Indicators* 126, 31–44. [https://doi.org/10.1016/S0031-0182\(96\)00068-5](https://doi.org/10.1016/S0031-0182(96)00068-5)
- Boers, N., 2018. Early-warning signals for Dansgaard-Oeschger events in a high-resolution ice core record. *Nat. Commun.* 9, 2556. <https://doi.org/10.1038/s41467-018-04881-7>
- Bourgon, N., Jaouen, K., Bacon, A.-M., Jochum, K.P., Dufour, E., Düringer, P., Ponche, J.-L., Joannes-Boyau, R., Boesch, Q., Antoine, P.-O., Hullot, M., Weis, U., Schulz-Kornas, E., Trost, M., Fiorillo, D., Demeter, F., Patole-Edoumba, E., Shackelford, L.L., Dunn, T.E., Zachwieja, A., Duangthongchit, S., Sayavonkhamdy, T., Sichanthongtip, P., Sihanam, D., Souksavatdy, V., Hublin, J.-J., Tütken, T., 2020. Zinc isotopes in Late Pleistocene fossil teeth from a Southeast Asian cave setting preserve paleodietary information. *Proc. Natl. Acad. Sci.* 117, 4675–4681. <https://doi.org/10.1073/pnas.1911744117>
- Britton, K., Gaudzinski-Windheuser, S., Roebroeks, W., Kindler, L., Richards, M.P., 2012. Stable isotope analysis of well-preserved 120,000-year-old herbivore bone collagen from the Middle Palaeolithic site of Neumark-Nord 2, Germany reveals niche separation between bovids and equids. *Palaeogeogr. Palaeoclimatol. Palaeoecol.* 333–334, 168–177. <https://doi.org/10.1016/j.palaeo.2012.03.028>
- Britton, K., Jimenez, E.-L., Le Corre, M., Pederzani, S., Daujeard, C., Jaouen, K., Vettese, D., Tütken, T., Hublin, J.-J., Moncel, M.-H., 2023. Multi-isotope zooarchaeological investigations at Abri du Maras: The paleoecological and paleoenvironmental context of Neanderthal subsistence strategies in the Rhône Valley during MIS 3. *J. Hum. Evol.* 174, 103292. <https://doi.org/10.1016/j.jhevol.2022.103292>
- Callaghan, T.V., Werkman, B.R., Crawford, Robert.M.M., 2002. The Tundra-Taiga Interface and Its Dynamics: Concepts and Applications. *Ambio* 6–14.
- Cao, Q.L., Pukazhenthil, B.S., Bapodra, P., Lowe, S., Bhatnagar, Y.V., 2023. Equid Adaptations to Cold Environments, in: Prins, H.H.T., Gordon, I.J. (Eds.), *The Equids: A Suite of Splendid Species, Fascinating Life Sciences*. Springer International Publishing, Cham, pp. 209–246. https://doi.org/10.1007/978-3-031-27144-1_8
- Drucker, D., Bocherens, H., 2004. Carbon and nitrogen stable isotopes as tracers of change in diet breadth during Middle and Upper Palaeolithic in Europe. *Int. J. Osteoarchaeol.* 14, 162–177. <https://doi.org/10.1002/oa.753>
- Drucker, D.G., 2022. The Isotopic Ecology of the Mammoth Steppe. *Annu. Rev. Earth Planet. Sci.* 50, 395–418. <https://doi.org/10.1146/annurev-earth-100821-081832>
- Drucker, D.G., Naito, Y.I., Coromina, N., Rufí, I., Soler, N., Soler, J., 2021. Stable isotope evidence of human diet in Mediterranean context during the Last Glacial Maximum. *J. Hum. Evol.* 154, 102967. <https://doi.org/10.1016/j.jhevol.2021.102967>
- Fabre, M., Lécuyer, C., Brugal, J.P., Amiot, R., Fourel, F., Martineau, F., 2011. Late Pleistocene climatic change in the French Jura (Gigny) recorded in the $\delta^{18}\text{O}$ of phosphate from

- ungulate tooth enamel. *Quat. Res.* 75, 605–613.
<https://doi.org/10.1016/j.yqres.2011.03.001>
- Harris, A.J.T., Elliott, D.A., 2019. Stable Isotope Studies of North American Arctic Populations: A Review 5, 11. <https://doi.org/10.5334/oq.67>
- Jaouen, K., Villalba-Mouco, V., Smith, G.M., Trost, M., Leichliter, J., Lüdecke, T., Méjean, P., Mandrou, S., Chmeleff, J., Guiserix, D., Bourgon, N., Blasco, F., Mendes Cardoso, J., Duquenoy, C., Moubtahij, Z., Salazar Garcia, D.C., Richards, M., Tütken, T., Hublin, J.-J., Utrilla, P., Montes, L., 2022. A Neandertal dietary conundrum: Insights provided by tooth enamel Zn isotopes from Gabasa, Spain. *Proc. Natl. Acad. Sci.* 119, e2109315119.
<https://doi.org/10.1073/pnas.2109315119>
- Jones, J.R., Britton, K., 2019. Multi-scale, integrated approaches to understanding the nature and impact of past environmental and climatic change in the archaeological record, and the role of isotope zooarchaeology. *J. Archaeol. Sci. Rep.* 23, 968–972.
<https://doi.org/10.1016/j.jasrep.2019.02.001>
- Julien, M.A., Bocherens, H., Burke, A., Drucker, D.G., Patou-Mathis, M., Krotova, O., Péan, S., 2012. Were European steppe bison migratory? 18O, 13C and Sr intra-tooth isotopic variations applied to a palaeoethological reconstruction. *Quat. Int.* 271, 106–119.
<https://doi.org/10.1016/j.quaint.2012.06.011>
- Kern, O.A., Koutsodendris, A., Allstädt, F.J., Mächtle, B., Peteet, D.M., Kalaitzidis, S., Christanis, K., Pross, J., 2022. A near-continuous record of climate and ecosystem variability in Central Europe during the past 130 kyrs (Marine Isotope Stages 5–1) from Füramoos, southern Germany. *Quat. Sci. Rev.* 284, 107505.
<https://doi.org/10.1016/j.quascirev.2022.107505>
- Kosintsev, P.A., Bachura, O.P., 2013. Late Pleistocene and Holocene mammal fauna of the Southern Urals. *Quat. Int., Quaternary interconnections in Eurasia: focus on Eastern Europe SEQS Conference, Rostov-on-Don, Russia, 21-26 June 2010* 284, 161–170.
<https://doi.org/10.1016/j.quaint.2012.06.022>
- Kovács, J., Moravcová, M., Újvári, G., Pintér, A.G., 2012. Reconstructing the paleoenvironment of East Central Europe in the Late Pleistocene using the oxygen and carbon isotopic signal of tooth in large mammal remains. *Quat. Int.* 276–277, 145–154.
<https://doi.org/10.1016/j.quaint.2012.04.009>
- Lécuyer, C., Hillaire-Marcel, C., Burke, A., Julien, M.-A., Hélie, J.-F., 2021. Temperature and precipitation regime in LGM human refugia of southwestern Europe inferred from $\delta^{13}\text{C}$ and $\delta^{18}\text{O}$ of large mammal remains. *Quat. Sci. Rev.* 255, 106796.
<https://doi.org/10.1016/j.quascirev.2021.106796>
- Lee-Thorp, J.A., 2008. On isotopes and old bones. *Archaeometry* 50, 925–950.
<https://doi.org/10.1111/j.1475-4754.2008.00441.x>
- Malikov, D.G., Svyatko, S.V., Pyryaev, A.N., 2023. Paleoecology of the mammoth fauna of Southern Siberia during the last glacial period based on stable isotope data. *Quat. Int.*
<https://doi.org/10.1016/j.quaint.2023.08.004>

- McCormack, J., Griffiths, M.L., Kim, S.L., Shimada, K., Karnes, M., Maisch, H., Pederzani, S., Bourgon, N., Jaouen, K., Becker, M.A., Jöns, N., Sisma-Ventura, G., Straube, N., Pollerspöck, J., Hublin, J.-J., Eagle, R.A., Tütken, T., 2022. Trophic position of *Otodus megalodon* and great white sharks through time revealed by zinc isotopes. *Nat. Commun.* 13, 2980. <https://doi.org/10.1038/s41467-022-30528-9>
- Murchie, T.J., Monteath, A.J., Mahony, M.E., Long, G.S., Cocker, S., Sadoway, T., Karpinski, E., Zazula, G., MacPhee, R.D.E., Froese, D., Poinar, H.N., 2021. Collapse of the mammoth-steppe in central Yukon as revealed by ancient environmental DNA. *Nat. Commun.* 12, 7120. <https://doi.org/10.1038/s41467-021-27439-6>
- Naito, Y.I., Chikaraishi, Y., Drucker, D.G., Ohkouchi, N., Semal, P., Wißing, C., Bocherens, H., 2016. Ecological niche of Neanderthals from Spy Cave revealed by nitrogen isotopes of individual amino acids in collagen. *J. Hum. Evol.* 93, 82–90. <https://doi.org/10.1016/j.jhevol.2016.01.009>
- Pederzani, S., Snoeck, C., Wacker, U., Britton, K., 2020. Anion exchange resin and slow precipitation preclude the need for pretreatments in silver phosphate preparation for oxygen isotope analysis of bioapatites. *Chem. Geol.* 534, 119455. <https://doi.org/10.1016/j.chemgeo.2019.119455>
- Peterson, B.J., Fry, B., 1987. Stable Isotopes in Ecosystem Studies. *Annu. Rev. Ecol. Syst.* 18, 293–320.
- Reade, H., Tripp, J.A., Charlton, S., Grimm, S.B., Leesch, D., Müller, W., Sayle, K.L., Fensome, A., Higham, T.F.G., Barnes, I., Stevens, R.E., 2020. Deglacial landscapes and the Late Upper Palaeolithic of Switzerland. *Quat. Sci. Rev.* 239. <https://doi.org/10.1016/j.quascirev.2020.106372>
- Richards, M.P., Taylor, G., Steele, T., McPherron, S.J.P., Soressi, M., Jaubert, J., Orschiedt, J., Mallye, J.B., Rendu, W., Hublin, J.-J., 2008. Isotopic dietary analysis of a Neanderthal and associated fauna from the site of Jonzac (Charente-Maritime), France. *J. Hum. Evol.* 55, 179–185. <https://doi.org/10.1016/j.jhevol.2008.02.007>
- Robinson, J.R., 2022. Investigating habitat heterogeneity of Late Pleistocene archaeological sites in eastern Africa from stable isotopes. *Hist. Biol.* 34, 674–693. <https://doi.org/10.1080/08912963.2021.1942465>
- Schwartz-Narbonne, R., Longstaffe, F.J., Kardynal, K.J., Druckenmiller, P., Hobson, K.A., Jass, C.N., Metcalfe, J.Z., Zazula, G., 2019. Reframing the mammoth steppe: Insights from analysis of isotopic niches. *Quat. Sci. Rev.* 215, 1–21. <https://doi.org/10.1016/j.quascirev.2019.04.025>
- Skrzypek, G., Winiewski, A., Grierson, P.F., 2011. How cold was it for Neanderthals moving to Central Europe during warm phases of the last glaciation? *Quat. Sci. Rev.* 30, 481–487. <https://doi.org/10.1016/j.quascirev.2010.12.018>
- Sponheimer, M., Robinson, T., Ayliffe, L.K., Roeder, B., Hammer, J., Passey, B.H., West, A., Cerling, T.E., Dearing, D., Ehleringer, J.R., 2003. Nitrogen isotopes in mammalian

- herbivores: hair d15N values from a controlled feeding study. *Int. J. Osteoarchaeol.* 13, 80–87. <https://doi.org/10.1002/oa.655>
- Stephan, E., 2017. Paläotemperaturbestimmungen anhand von Sauerstoffisotopenverhältnissen in Pferde und Rentierfunden aus der mittelpaläolithischen Fundstelle Salzgitter-Lebenstedt, Norddeutschland, in: Ludowici, B., Pöppelmann, H. (Eds.), *Die Tierknochenfunde Der Mittelpaläolithischen Jägerstation von Salzgitter-Lebenstedt, Forschungen Und Berichte Des Braunschweigischen Landesmuseums - Neue Folge, Band 1.* Braunschweigisches Landesmuseum.
- Stevens, R.E., Hedges, R.E.M., 2004. Carbon and nitrogen stable isotope analysis of northwest European horse bone and tooth collagen, 40,000 BP-present: Palaeoclimatic interpretations, in: *Quat. Sci. Rev.* pp. 977–991. <https://doi.org/10.1016/j.quascirev.2003.06.024>
- Szpak, P., 2023. Epilogue: Stable Isotope Analysis in Archaeology – Current Perspectives and Future Directions, in: Beasley, M.M., Somerville, A.D. (Eds.), *Exploring Human Behavior Through Isotope Analysis: Applications in Archaeological Research, Interdisciplinary Contributions to Archaeology.* Springer International Publishing, Cham, pp. 295–303. https://doi.org/10.1007/978-3-031-32268-6_13
- Wißing, C., Matzerath, S., Turner, E., Bocherens, H., 2015. Paleoecological and climatic implications of stable isotope results from late Pleistocene bone collagen, Ziegeleigrube Coenen, Germany. *Quat. Res. U. S.* 84, 96–105. <https://doi.org/10.1016/j.yqres.2015.05.005>
- Wißing, C., Rougier, H., Baumann, C., Comeyne, A., Crevecoeur, I., Drucker, D.G., Gaudzinski-Windheuser, S., Germonpré, M., Gómez-Olivencia, A., Krause, J., Matthies, T., Naito, Y.I., Posth, C., Semal, P., Street, M., Bocherens, H., 2019. Stable isotopes reveal patterns of diet and mobility in the last Neandertals and first modern humans in Europe. *Sci. Rep.* 9, 4433. <https://doi.org/10.1038/s41598-019-41033-3>
- Wißing, C., Rougier, H., Crevecoeur, I., Germonpré, M., Naito, Y.I., Semal, P., Bocherens, H., 2016. Isotopic evidence for dietary ecology of late Neandertals in North-Western Europe. *Quat. Int.*, 411, 327–345. <https://doi.org/10.1016/j.quaint.2015.09.091>
- Zimov, S.A., Zimov, N.S., Tikhonov, A.N., Chapin, F.S., 2012. Mammoth steppe: a high-productivity phenomenon. *Quat. Sci. Rev.* 57, 26–45. <https://doi.org/10.1016/j.quascirev.2012.10.005>

Decision Letter, first revision:

12th December 2023

38Dear Sarah,

Thank you for submitting your revised manuscript "Early *Homo sapiens* dispersed into cold steppes in central Europe" (NATECOLEVOL-23061429A). It has now been seen again by the original reviewers and their comments are below--note that the reason this is so late in getting to you is that we struggled to get in contact with reviewer 4 (Who also had remaining technical comments) so eventually I asked reviewer 2 to give them a check and they have signed off on them as well. The reviewers find that the paper has improved in revision, and therefore we'll be happy in principle to publish it in Nature Ecology & Evolution, pending minor revisions to satisfy the reviewers' final requests and to comply with our editorial and formatting guidelines.

We are now performing detailed checks on your paper and will send you a checklist detailing our editorial and formatting requirements in about a week. Please do not upload the final materials and make any revisions until you receive this additional information from us. I'm going to try and get this done as quickly as possible so that we don't delay publication, but in the meantime there are a few things you can probably get started with: the main text will need to be kept to 3500 words, the figures should not be included with the main manuscript file but uploaded separately, and I will be asking you to make the title more declarative (we should include the date, and a more obvious link to the other two papers would be good). You'll also need to complete updated versions of the editorial policy and reporting summary checklists--these would be good things to get going with if you need to make any changes to them.

[REDACTED]

Reviewer #2 (Remarks to the Author):

Dear Authors, dear Editor,

I have read the revised version of this manuscript with great interest and I am totally satisfied with the authors' responses. I would particularly like to thank them for their detailed and well-reasoned responses.

As a result, I am fully in favor of publishing this work, which should meet with great success in the relevant scientific community.

Congratulations to the authors on the quality of their work.

Prof. Dr. Christophe Lécuyer

39Our ref: NATECOLEVOL-23061429A

13th December 2023

Dear Dr. Pederzani,

Thank you for your patience as we've prepared the guidelines for final submission of your Nature Ecology & Evolution manuscript, "Early *Homo sapiens* dispersed into cold steppes in central Europe" (NATECOLEVOL-23061429A). Please carefully follow the step-by-step instructions provided in the attached file, and add a response in each row of the table to indicate the changes that you have made. Please also check and comment on any additional marked-up edits we have proposed within the text. Ensuring that each point is addressed will help to ensure that your revised manuscript can be swiftly handed over to our production team.

****We would like to start working on your revised paper, with all of the requested files and forms, as soon as possible (preferably within two weeks). Please get in contact with us immediately if you anticipate it taking more than two weeks to submit these revised files.****

In recognition of the time and expertise our reviewers provide to Nature Ecology & Evolution's editorial process, we would like to formally acknowledge their contribution to the external peer review of your manuscript entitled "Early *Homo sapiens* dispersed into cold steppes in central Europe". For those reviewers who give their assent, we will be publishing their names alongside the published article.

Nature Ecology & Evolution offers a Transparent Peer Review option for new original research manuscripts submitted after December 1st, 2019. As part of this initiative, we encourage our authors to support increased transparency into the peer review process by agreeing to have the reviewer comments, author rebuttal letters, and editorial decision letters published as a Supplementary item. When you submit your final files please clearly state in your cover letter whether or not you would like to participate in this initiative. Please note that failure to state your preference will result in delays in accepting your manuscript for publication.

40Cover suggestions

We welcome submissions of artwork for consideration for our cover. For more information, please see our [guide for cover artwork](https://www.nature.com/documents/Nature_covers_author_guide.pdf).

Nature Ecology & Evolution has now transitioned to a unified Rights Collection system which will allow our Author Services team to quickly and easily collect the rights and permissions required to publish your work. Approximately 10 days after your paper is formally accepted, you will receive an email in providing you with a link to complete the grant of rights. If your paper is eligible for Open Access, our Author Services team will also be in touch regarding any additional information that may be required to arrange payment for your article.

Please note that *Nature Ecology & Evolution* is a Transformative Journal (TJ). Authors may publish their research with us through the traditional subscription access route or make their paper immediately open access through payment of an article-processing charge (APC). Authors will not be required to make a final decision about access to their article until it has been accepted. [Find out more about Transformative Journals](https://www.springernature.com/gp/open-research/transformative-journals)

Authors may need to take specific actions to achieve [compliance with funder and institutional open access mandates](https://www.springernature.com/gp/open-research/funding/policy-compliance-faqs). If your research is supported by a funder that requires immediate open access (e.g. according to [Plan S principles](https://www.springernature.com/gp/open-research/plan-s-compliance)) then you should select the gold OA route, and we will direct you to the compliant route where possible. For authors selecting the subscription publication route, the journal's standard licensing terms will need to be accepted, including [self-archiving-and-license-to-publish](https://www.nature.com/nature-portfolio/editorial-policies/self-archiving-and-license-to-publish). Those licensing terms will supersede any other terms that the author or any third party may assert apply to any version of the manuscript.

[REDACTED]

[REDACTED]

Reviewer #2:

Remarks to the Author:

Dear Authors, dear Editor,

I have read the revised version of this manuscript with great interest and I am totally satisfied with the authors' responses. I would particularly like to thank them for their detailed and well-reasoned responses.

As a result, I am fully in favor of publishing this work, which should meet with great success in the relevant scientific community.

Congratulations to the authors on the quality of their work.

Prof. Dr. Christophe Lécuyer

Final Decision Letter:

19th December 2023

Dear Dr Pederzani,

I am writing in the temporary absence of my colleague Luiseach Nic Eoin.

We are pleased to inform you that your Article entitled "Stable isotopes show *Homo sapiens* dispersed into cold steppes ~45,000 years ago at Ilsenhöhle in Ranis, Germany", has now been accepted for publication in Nature Ecology & Evolution.

Over the next few weeks, your paper will be copyedited to ensure that it conforms to Nature Ecology and Evolution style. Once your paper is typeset, you will receive an email with a link to choose the appropriate publishing options for your paper and our Author Services team will be in touch regarding any additional information that may be required

Due to the importance of these deadlines, we ask you please us know now whether you will be difficult to contact over the next month. If this is the case, we ask you provide us with the contact information

42(email, phone and fax) of someone who will be able to check the proofs on your behalf, and who will be available to address any last-minute problems. Once your paper has been scheduled for online publication, the Nature press office will be in touch to confirm the details.

Acceptance of your manuscript is conditional on all authors' agreement with our publication policies (see www.nature.com/authors/policies/index.html). In particular your manuscript must not be published elsewhere and there must be no announcement of the work to any media outlet until the publication date (the day on which it is uploaded onto our web site).

Please note that *Nature Ecology & Evolution* is a Transformative Journal (TJ). Authors may publish their research with us through the traditional subscription access route or make their paper immediately open access through payment of an article-processing charge (APC). Authors will not be required to make a final decision about access to their article until it has been accepted. [Find out more about Transformative Journals](https://www.springernature.com/gp/open-research/transformative-journals)

Authors may need to take specific actions to achieve [compliance with funder and institutional open access mandates](https://www.springernature.com/gp/open-research/funding/policy-compliance-faqs). If your research is supported by a funder that requires immediate open access (e.g. according to [Plan S principles](https://www.springernature.com/gp/open-research/plan-s-compliance)) then you should select the gold OA route, and we will direct you to the compliant route where possible. For authors selecting the subscription publication route, the journal's standard licensing terms will need to be accepted, including [those licensing terms](https://www.nature.com/nature-portfolio/editorial-policies/self-archiving-and-license-to-publish) will supersede any other terms that the author or any third party may assert apply to any version of the manuscript.

We welcome the submission of potential cover material (including a short caption of around 40 words) related to your manuscript; suggestions should be sent to Nature Ecology & Evolution as electronic files (the image should be 300 dpi at 210 x 297 mm in either TIFF or JPEG format). Please note that such pictures should be selected more for their aesthetic appeal than for their scientific content, and that colour images work better than black and white or grayscale images. Please do not try to design a

43cover with the Nature Ecology & Evolution logo etc., and please do not submit composites of images related to your work. I am sure you will understand that we cannot make any promise as to whether any of your suggestions might be selected for the cover of the journal.

You can generate the link yourself when you receive your article DOI by entering it here: <http://authors.springernature.com/share>.

[REDACTED]

P.S. Click on the following link if you would like to recommend Nature Ecology & Evolution to your librarian <http://www.nature.com/subscriptions/recommend.html#forms>

** Visit the Springer Nature Editorial and Publishing website at http://editorial-jobs.springernature.com?utm_source=ejp_NEcoE_email&utm_medium=ejp_NEcoE_email&utm_campaign=ejp_NEcoE for more information about our career opportunities. If you have any questions please click [here](mailto:editorial.publishing.jobs@springernature.com). **